# PROVABLE SHARPNESS-AWARE MINIMIZATION WITH ADAPTIVE LEARNING RATE

## ABSTRACT

Sharpness aware minimization (SAM) optimizer has been extensively explored as it can converge fast and train deep neural networks efficiently via introducing extra perturbation steps to flatten the landscape of deep learning models. A combination of SAM with adaptive learning rate (AdaSAM) has also been explored to train large-scale deep neural networks without theoretical guarantee due to the dual difficulties in analyzing the perturbation step and the coupled adaptive learning rate. In this paper, we try to analyze the convergence rate of AdaSAM in the stochastic non-convex setting. We theoretically show that AdaSAM admit a $\mathcal{O}(1/\sqrt{bT})$ convergence rate and show linear speedup property with respect to mini-batch size $b$. To the best of our knowledge, we are the first to provide the non-trivial convergence rate of SAM with an adaptive learning rate. To decouple the two stochastic gradient steps with the adaptive learning rate, we introduce the delayed second-order momentum term in the convergence to decompose them to make them independent while taking an expectation. Then we bound them by showing the adaptive learning rate has a limited range, which makes our analysis feasible. At last, we conduct experiments on several NLP tasks and they show that AdaSAM could achieve superior performance compared with SGD, AMSGrad, and SAM optimizer.

## 1 INTRODUCTION

Sharpness-aware minimization (SAM) Foret et al. (2021) is a powerful optimizer for training large-scale deep learning models by minimizing the gap between the training performance and generalization performance. It has achieved remarkable results on various deep neural networks, such as ResNet He et al. (2016), vision transformer Dosovitskiy et al. (2021); Xu et al. (2021), language models Devlin et al. (2018); He et al. (2020), on extensive benchmarks.

However, SAM-type methods suffer from several issues during training the deep neural network, especially for huge computation costs and heavily hyper-parameter tuning procedure. It needs double gradients computation compared with classic optimizers, like SGD, AdamKingma & Ba (2015), AMSGrad Reddi et al. (2018), due to the extra perturbation step. Hence, SAM requires to forward and back propagate twice for one parameter update, resulting in one more computation cost than the classic optimizers. Moreover, as there are two steps during the training process, it needs double hyper-parameters, which makes the learning rate tuning unbearable and costly.

Adaptive learning rate optimization methods scale the gradients based on the history gradient information to accelerate convergence. These methods, such as Adagrad Duchi et al. (2011), Adam, and AMSGrad, have been studied for solving the NLP tasks Zhang et al. (2020). However, it might converge to a local minima which leads to poor generalization performance. Many studies have investigated how to improve the generalization ability of the adaptive learning rate methods. Recently, several work has tried to ease the learning rate tuning in SAM by combining SAM with adaptive learning rate. For example, Zhuang et al. (2022) trains ViT models with an adaptive method and Kwon et al. (2021) training a NLP model. Although remarkable performance has been achieved, their convergence is still unknown since the adaptive learning rate is used in SAM. Directly analyzing the convergence of this method is difficult due to the two steps of optimization, especially the adaptive learning rate adopted in the second optimization step.

In this paper, we analyze the convergence rate of SAM with an adaptive learning rate in the non-convex stochastic setting, dubbed AdaSAM. To circumvent the difficulty in the analysis, we develop technique to decouple the two-step training of SAM and the adaptive learning rate. The analysis procedure is mainly divided into two parts. The first part is to analyze the procedure of the SAM. Then we analyze the second step that adopts the adaptive learning rate method. We introduce a second-order momentum term from the previous iteration, which is related to the adaptive learning rate and independent of SAM while taking an expectation. Then we can bound the term composed by the SAM and the previous second-order momentum due to the limited adaptive learning rate. We prove that AdaSAM enjoys the property of linear speedup with respect to the batch size. Furthermore, we apply AdaSAM to train RoBERTa model on the GLUE benchmark. We show that AdaSAM achieves the best performance in experiments, where it wins 6 tasks of 8 tasks, and the linear speedup can be clearly observed.

In the end, we summarize our contributions as follows:

- We present the first convergence guarantee of the adaptive SAM method under the non-convex setting. Our results suggest that a large mini-batch can help convergence due to the linear speedup with respect to batch size.
- We conduct a series of experiments on various tasks. The results show that AdaSAM outperforms the most of state-of-art optimizers and the linear speedup is verified.

## 2 PRELIMINARY

### 2.1 PROBLEM SETUP

In this work, we focus on the following stochastic nonconvex optimization

$$\min_{x \in \mathbb{R}^d} f(x) := \mathbb{E}_{\xi \sim D} f_\xi(x), \tag{1}$$

where $d$ is dimension of variable $x$, $D$ is the unknown distribution of the data samples, $f_\xi(x)$ is a smooth and possibly non-convex function, and $f_{\xi_i}(x)$ denotes the objective function at the sampled data point $\xi_i$ according to data distribution $D$. In machine learning, it covers empirical risk minimization as a special case and $f$ is the loss function. When the dataset $D$ cover $N$ data points, i.e., $D = \{\xi_i, i = 1, 2, \ldots, N\}$. Problem 1 reduces to the following finite-sum problem:

$$\min_{x \in \mathbb{R}^d} f(x) := \frac{1}{N} \sum_i f_{\xi_i}(x). \tag{2}$$

Without additional declaration, we represent $f_i(x)$ as $f_{\xi_i}(x)$ for simplification.

**Notations.** $f_i(x)$ is the $i$-th loss function while $x \in \mathbb{R}^d$ is the model parameter and $d$ is the parameter dimension. We denote the $l_2$ norm as $\| \cdot \|_2$. A Hadamard product is denoted as $a \odot b$ where $a,b$ are two vectors. For a vector $a \in \mathbb{R}^d$, $\sqrt{a}$ is denoted as a vector that the $j$-th value, $(\sqrt{a})_{(j)}$, is equal to the square root of $a_j$

### 2.2 RELATED WORK

**Sharpness-aware minimization** Previous work shows that sharp minima may lead to poor generalization whereas flat minima perform betterJiang et al. (2020); Keskar et al. (2017); He et al. (2019). Therefore, it is popular to consider sharpness to be closely related to the generalization. Sharpness-aware minimization is proposed in Foret et al. (2021), which targets to find flat minimizers by minimizing the training loss uniformly in entire neighborhood. Specifically, Foret et al. (2021) aims to solve the following minimax saddle point problem:

$$\min_x \max_{\|\delta\| \leq \rho} f(x + \delta) + \lambda \|x\|_2^2, \tag{3}$$

where $\rho \geq 0$ and $\lambda \geq 0$ are two hyperparameters. That is, the perturbed loss function of $f(x)$ in a neighborhood is minimized instead of the original loss function $f(x)$. By using Taylor expansion of $f(x + \delta)$ with respect to $\delta$, the inner max problem is approximately solved via

$$\delta^*(x) = \arg\max_{\|\delta\| \leq \rho} f(x + \delta) \approx \arg\max_{\|\delta\| \leq \rho} f(x) + \delta^\top \nabla f(x) = \arg\max_{\|\delta\| \leq \rho} \delta^\top \nabla f(x) = \rho \frac{\nabla f(x)}{\|\nabla f(x)\|}.$$

Dropping the quadratic term, (3) is simplified as the following minimization problem

$$\min_x f\left(x + \rho \frac{\nabla f(x)}{\|\nabla f(x)\|}\right). \tag{4}$$

The stochastic gradient of $f\left(x + \rho \frac{\nabla f(x)}{\|\nabla f(x)\|}\right)$ on a batch data $b$ includes the Hessian-vector product, SAM further approximates the gradient by

$$\nabla_x f_b\left(x + \rho \frac{\nabla f_b(x)}{\|\nabla f_b(x)\|}\right) \approx \nabla_x f_b(x)\big|_{x + \rho \frac{\nabla f_b(x)}{\|\nabla f_b(x)\|}}.$$

Then, along the negative direction $-\nabla_x f_b(x)\big|_{x + \rho \frac{\nabla f_b(x)}{\|\nabla f_b(x)\|}}$, SGD is applied to solve minimization problem (4). It is easy to see that SAM requires twice gradient back-propagation, i.e., $\nabla f_b(x)$ and $\nabla_x f_b(x)\big|_{x + \rho \frac{\nabla f_b(x)}{\|\nabla f_b(x)\|}}$. Due to the existence of hyperparameter $\rho$, one needs to carefully tune both $\rho$ and learning rate in SAM. In practice, $\rho$ is predefined to control the radius of the neighborhood.

Some variants of SAM are proposed to improve its performance. Recently, several works Zhuang et al. (2022); Kwon et al. (2021) have empirically incorporated adaptive learning rate with SAM and shown impressive generalization accuracy, while their convergence analysis has never been studied. ESAM Du et al. (2022) proposes an efficient method by sparsifying the gradients will alleviate the double computation cost of backpropagation. ASAM Kwon et al. (2021) modifies SAM by adaptively scaling the neighborhood so that the sharpness is invariant to parameters re-scaling. GSAM Zhuang et al. (2022) simultaneously minimizes the perturbed function and a new defined surrogate gap function to further improve the flatness of minimizers. Liu et al. Liu et al. (2022) also studies SAM in large-batch training scenario and periodically update the perturbed gradient. On the other hand, there are some works analyzing the convergence of the SAM such as Andriushchenko & Flammarion (2022), where it does not consider a normalization step, i.e., the normalization in $\frac{\nabla f_b(x)}{\|\nabla f_b(x)\|}$.

**Adaptive optimizer**  The adaptive learning rate method can automatically adjust the learning rate based on the history gradients methods. The first adaptive method is Adagrad Duchi et al. (2011), and it can achieve a better result than other first-order methods under the convex setting. While training the deep neural network, Adagrad will decrease the learning rate rapidly and the result is not good. Adadelta Zeiler (2012) is proposed to change this situation and introduces a learning rate based on the exponential average history gradients. Adam Kingma & Ba (2015) adds momentum, and it shows great performance in many tasks. However, Reddi et al Reddi et al. (2018) give an example that it cannot converge even when the objective function is convex. They propose a new method called AMSGrad that can solve this method. Many papers also analyze the convergence of the adaptive methods Zhou et al. (2018); Chen et al. (2019); Zaheer et al. (2018); Ward et al. (2019); Défossez et al. (2020); Zou et al. (2019); Chen et al. (2021).

## 3   ADASAM: SAM WITH ADAPTIVE LEARNING RATE

In this section, we introduce SAM with the adaptive learning rate (AdaSAM) from the AMSGrad optimizer. Then, we will present the convergence results of AdaSAM. At last, we give the proof sketch for main theorem.

AdaSAM is described as in Algorithm 1. In each iteration, a mini-batch gradient estimation $g_t$ at point $x + \epsilon(x)$ with batchsize $b$ is computed, i.e.,

$$g_t = \nabla_x f_b(x)\big|_{x_t + \epsilon(x_t)} = \frac{1}{b}\sum_{i \in B} \nabla f_{\xi_i}(x_t + \delta(x_t)).$$

Here, $\delta(x_t)$ is the extra perturbed gradient step in SAM that is given as follows

$$\delta(x_t) = \rho \frac{s_t}{\|s_t\|}, \text{ where } s_t = \nabla_x f_b(x)\big|_{x_t} = \frac{1}{b}\sum_{i \in B} \nabla f_{\xi_i}(x_t).$$

Then, exponential averaging of $g_t$ and the second-order term $[g_t]^2$ are accumulatively computed as $m_t$ and $v_t$, respectively. AdaSAM then updates iterate along $-m_t$ with the adaptive learning rate $\gamma \eta_t$.

---

**Algorithm 1:** AdaSAM: SAM with adaptive learning rate

---

**Input:** Initial parameters $x_0$, $m_{-1} = 0$, $\hat{v}_{-1} = \epsilon^2$ (a small positive scalar to avoid the denominator diminishing), base learning rate $\gamma$, neighborhood size $\rho$ and momentum parameters $\beta_1$, $\beta_2$.

**Output:** Optimized parameter $x_{T+1}$

1 **for** *iteration* $t \in \{0, 1, 2, ..., T-1\}$ **do**
2     Sample mini-batch $B = \{\xi_{t_1}, \xi_{t_2}, ..., \xi_{t_{|B|}}\}$;
3     Compute gradient $s_t = \nabla_x f_B(x)|_{x_t} = \frac{1}{b} \sum_{i \in B} \nabla f_{t_i}(x_t)$;
4     Compute $\delta(x_t) = \rho_t \frac{s_t}{\|s_t\|}$;
5     Compute SAM gradient $g_t = \nabla_x f_B(x)|_{x_t + \delta(x_t)}$;
6     $m_t = \beta_1 m_{t-1} + (1 - \beta_1) g_t$;
7     $v_t = \beta_2 v_{t-1} + (1 - \beta_2)[g_t]^2$;
8     $\hat{v}_t = \max(\hat{v}_{t-1}, v_t)$;
9     $\eta_t = 1/\sqrt{\hat{v}_t}$;
10     $x_{t+1} = x_t - \gamma m_t \odot \eta_t$;
11 **end**

---

*Remark* 3.1. Below, we give several comments on AdaSAM:

- When $\beta_2 = 1$, the adaptive learning rate reduce to the diminishing one as SGD. Then, AdaSAM recovers the classic SAM optimizer.

- If we drop out the 8-th line $\hat{v}_t = \max(\hat{v}_{t-1}, v_t)$, then our algorithm becomes the variant of Adam. The counterexample that Adam does not converge in the Reddi et al. (2018) also holds for the SAM variant, while AdaSAM can converge.

### 3.1 CONVERGENCE ANALYSIS

Before stating our convergence analysis, we first introduce some useful assumptions.

**Assumption 3.2.** *L*-**smooth**. $f_i$ and $f$ is differentiable with gradient Lipschitz property:

$$\|\nabla f_i(x) - \nabla f_i(y)\| \le L\|x - y\|, \|\nabla f(x) - \nabla f(y)\| \le L\|x - y\|, \forall x, y \in \mathbb{R}^d, i = 1, 2, ..., N,$$

which also implies the descent inequality, i.e., $f_i(y) \le f_i(x) + \langle \nabla f_i(x), y - x \rangle + \frac{L}{2}\|y - x\|^2$.

**Assumption 3.3. Bounded variance**. The estimator of the gradient is unbiased and the variance of the stochastic gradient is bounded:

$$\mathbb{E}\nabla f_i(x) = \nabla f(x), \quad \mathbb{E}\|\nabla f_i(x) - \nabla f(x)\|^2 \le \sigma^2.$$

When the mini-batch size $b$ is used, we have $\mathbb{E}\|\nabla f_b(x) - \nabla f(x)\|^2 \le \frac{\sigma^2}{b}$.

**Assumption 3.4. Bounded stochastic gradients**. The stochastic gradient is uniformly bounded, *i.e.*,

$$\|\nabla f_i(x)\|_\infty \le G, \, for \, any \, i = 1, \dots, N.$$

*Remark* 3.5. The above assumptions are commonly used in the proof of the convergence for adaptive stochastic gradient methods such as Cutkosky & Orabona (2019); Huang et al. (2021); Zhou et al. (2018); Chen et al. (2019).

Note that AdaSAM is the combination of AMSGrad and SAM. We briefly explain the main idea of analysis in AMSGradReddi et al. (2018) and SAMAndriushchenko & Flammarion (2022).

We firstly notice that the main step in AMSGrad analysis is to estimate the expectation $\mathbb{E}[x_{t+1} - x_t] = -\mathbb{E}m_t \odot \eta_t = -\mathbb{E}(1 - \beta_1)g_t \odot \eta_t - \mathbb{E}\beta_1 m_{t-1} \odot \eta_t$, which is conditioned on the filtration $\sigma(x_t)$. First, we consider the situation that $\beta_1 = 0$ which does not include the momentum. Some works Zaheer et al. (2018); Savarese et al. (2021) apply delay technology to split the dependence between $g_t$ and $\eta_t$, that is

$$\mathbb{E}g_t \odot \eta_t = \mathbb{E}[g_t \odot \eta_{t-1}] + \mathbb{E}[g_t \odot (\eta_t - \eta_{t-1})] = \nabla f(x_t) \odot \eta_{t-1} + \mathbb{E}[g_t \odot (\eta_t - \eta_{t-1})].$$

The second term $\mathbb{E}[g_t \odot (\eta_t - \eta_{t-1})]$ is dominated by the first term $\nabla f(x_t) \odot \eta_{t-1}$. Then, it is not difficult to get the convergence result of AMSGrad. When we apply the same strategy to AdaSAM, we

find that $\mathbb{E}g_t \odot \eta_{t-1}$ cannot be handled similarly because $\mathbb{E}g_t = \mathbb{E}\nabla_x f_b\left(x + \rho\frac{\nabla f_b(x)}{\|\nabla f_b(x)\|}\right) \neq \nabla f(x_t)$. Inspired by (Andriushchenko & Flammarion, 2022, Lemma 16), our key observation is that

$$\mathbb{E}\nabla_x f_b\left(x + \rho\frac{\nabla f_b(x)}{\|\nabla f_b(x)\|}\right) \approx \mathbb{E}\nabla_x f_b\left(x + \rho\frac{\nabla f(x)}{\|\nabla f(x)\|}\right) = \nabla_x f\left(x + \rho\frac{\nabla f(x)}{\|\nabla f(x)\|}\right)$$

and we prove the other terms such as $\mathbb{E}\left(\nabla_x f_b\left(x + \rho\frac{\nabla f_b(x)}{\|\nabla f_b(x)\|}\right) - \nabla_x f_b\left(x + \rho\frac{\nabla f(x)}{\|\nabla f(x)\|}\right)\right) \odot \eta_{t-1}$ have small values that do not dominate the convergence rate.

Then, when we apply the momentum method, we find that the term $\mathbb{E}m_{t-1} \odot \eta_t$ cannot be ignored. By introducing an auxiliary sequence $z_t = x_t + \frac{\beta_1}{1-\beta_1}(x_t - x_{t-1})$, we have $\mathbb{E}[z_{t+1} - z_t] = -\mathbb{E}[\frac{\beta_1}{1-\beta_1}\gamma m_{t-1} \odot (\eta_{t-1} - \eta_t) - \gamma g_t \odot \eta_t]$. The first term contains the momentum term which has a small value and we can remove it without hurting the convergence rate.

**Theorem 3.6.** *Under the assumptions 3.2,3.3,3.4, and $\gamma$ is a fixed number that satisfies that $\gamma \leq \frac{\epsilon}{16L}$, for the sequence $\{x_t\}$ generated by Algorithm 1, we have the following convergence rate*

$$\frac{1}{T}\sum_{t=0}^{T-1}\mathbb{E}\|\nabla f(x_t)\|_2^2 \leq \frac{2G(f(x_0) - f^*)}{\gamma T} + \frac{8G\gamma L}{\epsilon}\frac{\sigma^2}{b\epsilon} + \Phi \tag{5}$$

*where*

$$\Phi = \frac{45GL^2\rho_t^2}{\epsilon} + \frac{2G^3}{(1-\beta_1)T}d(\frac{1}{\epsilon} - \frac{1}{G}) + \frac{8G\gamma L}{\epsilon}\frac{L\rho_t^2}{\epsilon}$$
$$+ \frac{2(4 + (\frac{\beta_1}{1-\beta_1})^2)\gamma LG^3}{T}d(\epsilon^{-2} - G^{-2}) + \frac{6\gamma^2 L^2\beta_1^2}{(1-\beta_1)^2}\frac{dG^3}{\epsilon^3}. \tag{6}$$

**Corollary 3.7** (mini-batch linear speedup). *Under the same conditions of Theorem 3.6. Furthermore, when we choose the learning rate $\gamma = O(\sqrt{\frac{b}{T}})$ and neighborhood size $\rho = O(\sqrt{\frac{1}{bT}})$, we have*

$$\frac{1}{T}\sum_{t=0}^{T-1}\mathbb{E}\|\nabla f(x_t)\|_2^2 = O\left(\frac{1}{\sqrt{bT}}\right) + O\left(\frac{1}{bT}\right) + O\left(\frac{1}{T}\right) + O\left(\frac{1}{b^{\frac{1}{2}}T^{\frac{3}{2}}}\right) + O\left(\frac{b^{\frac{1}{2}}}{T^{\frac{3}{2}}}\right) + O\left(\frac{b}{T}\right).$$

*When $T$ is sufficiently large, we achieve the linear speedup convergence rate with respect to mini-batch size $b$, i.e.,*

$$\frac{1}{T}\sum_{t=0}^{T-1}\mathbb{E}\|\nabla f(x_t)\|_2^2 = O\left(\frac{1}{\sqrt{bT}}\right). \tag{7}$$

*Remark* 3.8. To reach a $O(\delta)$ stationary point, when the batch size is 1, it needs $T = O(\frac{1}{\delta^2})$ iterations. When the batch size is $b$, we need to run $T = O(\frac{1}{b\delta^2})$ steps. The method with batch size $b$ is $b$ times faster than batch size of 1.

### 3.2 PROOF SKETCH

In this part, we will give the proof sketch of the Theorem 3.6. For the complete proof, please see Appendix. We first introduce an auxiliary sequence $z_t = x_t + \frac{\beta_1}{1-\beta_1}(x_t - x_{t-1})$. By applying $L$-smooth condition, we have

$$f(z_{t+1}) \leq f(z_t) + \langle\nabla f(z_t), z_{t+1} - z_t\rangle + \frac{L}{2}\|z_{t+1} - z_t\|^2. \tag{8}$$

Applying it to the sequence $\{z_t\}$ and using the delay strategy yield

$$f(z_{t+1}) - f(z_t) \leq \langle\nabla f(z_t), \frac{\gamma\beta_1}{1-\beta_1}m_{t-1} \odot (\eta_{t-1} - \eta_t)\rangle + \frac{L}{2}\|z_{t+1} - z_t\|^2$$
$$+ \langle\nabla f(z_t), \frac{\gamma}{b}\sum_{i \in B}\nabla f_i(x_t + \rho_t\frac{s_t}{\|s_t\|}) \odot (\eta_{t-1} - \eta_t)\rangle$$

$$+ \langle \nabla f(z_t) - \nabla f(x_t), -\frac{\gamma}{b} \sum_{i \in B} \nabla f_i(x_t + \rho_t \frac{s_t}{\|s_t\|}) \odot \eta_{t-1} \rangle$$

$$+ \langle \nabla f(x_t), \frac{\gamma}{b} \sum_{i \in B} \nabla f_i(x_t + \rho_t \frac{\nabla f(x_t)}{\|\nabla f(x_t)\|}) \odot \eta_{t-1} - \frac{\gamma}{b} \sum_{i \in B} \nabla f_i(x_t + \rho_t \frac{s_t}{\|s_t\|}) \odot \eta_{t-1} \rangle$$

$$+ \langle \nabla f(x_t), -\frac{\gamma}{b} \sum_{i \in B} \nabla f_i(x_t + \rho_t \frac{\nabla f(x_t)}{\|\nabla f(x_t)\|}) \odot \eta_{t-1} \rangle. \tag{9}$$

From the Lemma B.5, Lemma B.6, Lemma B.7 in appendix, we can bound the above terms in (9) as follows

$$\langle \nabla f(z_t), \frac{\gamma}{b} \sum_{i \in B} \nabla f_i(x_t + \rho_t \frac{s_t}{\|s_t\|}) \odot (\eta_{t-1} - \eta_t) \rangle \le \gamma G^2 \|\eta_{t-1} - \eta_t\|_1 \tag{10}$$

$$\langle \nabla f(z_t), \frac{\gamma \beta_1}{1 - \beta_1} m_{t-1} \odot (\eta_{t-1} - \eta_t) \rangle \le \frac{\gamma \beta_1}{1 - \beta_1} G^2 \|\eta_{t-1} - \eta_t\|_1 \tag{11}$$

$$\langle \nabla f(x_t), \frac{\gamma}{b} \sum_{i \in B} \nabla f_i(x_t + \rho_t \frac{\nabla f(x_t)}{\|\nabla f(x_t)\|}) \odot \eta_{t-1} - \frac{\gamma}{b} \sum_{i \in B} \nabla f_i(x_t + \rho_t \frac{s_t}{\|s_t\|}) \odot \eta_{t-1} \rangle$$

$$\le \frac{\gamma}{2\mu^2} \|\nabla f(x_t) \odot \sqrt{\eta_{t-1}}\|^2 + \frac{2\mu^2 \gamma L^2 \rho_t^2}{\epsilon}. \tag{12}$$

Then we substitute them into the (9), and take the conditional expectation to get

$$\mathbb{E} f(z_{t+1}) - f(z_t) \le \mathbb{E} \langle \nabla f(x_t), -\frac{\gamma}{b} \sum_{i \in B} \nabla f_i(x_t + \rho_t \frac{\nabla f(x_t)}{\|\nabla f(x_t)\|}) \odot \eta_{t-1} \rangle + \frac{L}{2} \mathbb{E} \|z_{t+1} - z_t\|^2$$

$$+ \frac{\gamma}{2\mu^2} \|\nabla f(x_t) \odot \sqrt{\eta_{t-1}}\|^2 + \frac{2\mu^2 \gamma L^2 \rho_t^2}{\epsilon} + \frac{\gamma}{1 - \beta_1} G^2 \|\eta_{t-1} - \eta_t\|_1$$

$$+ \mathbb{E} \langle \nabla f(z_t) - \nabla f(x_t), -\frac{\gamma}{b} \sum_{i \in B} \nabla f_i(x_t + \rho_t \frac{s_t}{\|s_t\|}) \odot \eta_{t-1} \rangle, \tag{13}$$

where $\mu > 0$ is to be determined.

From the Lemma B.8, Lemma B.10 and Lemma B.9 in Appendix, we have

$$\mathbb{E} \langle \nabla f(x_t), -\frac{\gamma}{b} \sum_{i \in B} \nabla f_i(x_t + \rho_t \frac{\nabla f(x_t)}{\|\nabla f(x_t)\|}) \odot \eta_{t-1} \rangle$$

$$\le -\gamma \|\nabla f(x_t) \odot \sqrt{\eta_{t-1}}\|^2 + \mathbb{E} \frac{\gamma}{2\alpha^2} \|\nabla f(x_t) \odot \sqrt{\eta_{t-1}}\|^2 + \frac{\gamma \alpha^2 L^2 \rho_t^2}{2\epsilon} \tag{14}$$

$$\frac{L}{2} \mathbb{E} \|z_{t+1} - z_t\|^2 \le \frac{LG^2 \gamma^2 \beta_1^2}{(1 - \beta_1)^2} \mathbb{E} \|\eta_t - \eta_{t-1}\|^2 + \gamma^2 L (3 \frac{1 + \beta}{\beta \epsilon} (\mathbb{E} \|\nabla f(x_t) \odot \sqrt{\eta_{t-1}}\|^2 + \frac{L \rho_t^2}{\epsilon} + \frac{\sigma^2}{b\epsilon})$$

$$+ (1 + \beta) G^2 \mathbb{E} \|\eta_t - \eta_{t-1}\|^2) \tag{15}$$

$$\mathbb{E} \langle \nabla f(z_t) - \nabla f(x_t), -\frac{\gamma}{b} \sum_{i \in B} \nabla f_i(x_t + \rho_t \frac{s_t}{\|s_t\|}) \odot \eta_{t-1} \rangle$$

$$\le \frac{\gamma^3 L^2 \beta_1^2}{2\epsilon (1 - \beta_1)^2} (\frac{1}{\lambda_1^2} + \frac{1}{\lambda_2^2} + \frac{1}{\lambda_3^2}) \frac{dG_\infty^2}{\epsilon^2} + \frac{\gamma \lambda_1^2}{2} \|\nabla f(x_t) \odot \sqrt{\eta_{t-1}}\|^2 + \frac{\gamma L^2 \rho_t^2}{2\epsilon} (\lambda_2^2 + 4\lambda_3^2). \tag{16}$$

Next, we substitute it into the (13). Taking the expectation over all history information yields

$$\mathbb{E} f(x_{t+1}) - \mathbb{E} f(x_t) \le -\gamma (1 - \frac{1}{2\mu^2} - \frac{1}{2\alpha^2} - \frac{3\gamma L(1 + \beta)}{\beta \epsilon} - \frac{\lambda_1^2}{2}) \mathbb{E} \|\nabla f(x_t) \odot \sqrt{\eta_{t-1}}\|^2$$

$$+ \frac{2\mu^2 \gamma L^2 \rho_t^2}{\epsilon} + \frac{\gamma}{1 - \beta_1} G^2 \mathbb{E} \|\eta_{t-1} - \eta_t\|_1 + \frac{\gamma^3 L^2 \beta_1^2}{2\epsilon (1 - \beta_1)^2} (\frac{1}{\lambda_1^2} + \frac{1}{\lambda_2^2} + \frac{1}{\lambda_3^2}) \frac{dG_\infty^2}{\epsilon^2} + \frac{\gamma L^2 \rho_t^2}{2\epsilon} (\lambda_2^2 + 4\lambda_3^2)$$

$$+ \frac{\gamma \alpha^2 L^2 \rho^2}{2\epsilon} + \frac{3\gamma^2 L(1 + \beta)}{\beta \epsilon} (\frac{L \rho_t^2}{\epsilon} + \frac{\sigma^2}{b\epsilon}) + \gamma^2 LG^2 ((\frac{\beta_1}{1 - \beta_1})^2 + 1 + \beta) \mathbb{E} \|\eta_t - \eta_{t-1}\|^2. \tag{17}$$

We set $\mu^2 = \alpha^2 = 8$, $\beta = 3$, $\lambda_1^2 = \frac{1}{4}$, $\lambda_2^2 = \lambda_3^2 = 1$ and we choose $\frac{2\gamma L}{\epsilon} \leq \frac{1}{8}$. Note that $\eta_t$ is bounded. We have

$$\frac{\gamma}{2G}\mathbb{E}\|\nabla f(x_t)\|^2 \leq \frac{\gamma}{2}\mathbb{E}\|\nabla f(x_t) \odot \sqrt{\eta_{t-1}}\|^2 \tag{18}$$

$$\leq -\mathbb{E}f(x_{t+1}) + \mathbb{E}f(x_t) + \frac{45\gamma L^2 \rho_t^2}{2\epsilon} + \frac{\gamma}{1-\beta_1}G^2\mathbb{E}\|\eta_{t-1} - \eta_t\|_1$$

$$+ \frac{4\gamma^2 L}{\epsilon}\left(\frac{L\rho_t^2}{\epsilon} + \frac{\sigma^2}{b\epsilon}\right) + \left(4 + \left(\frac{\beta_1}{1-\beta_1}\right)^2\right)\gamma^2 LG^2\mathbb{E}\|\eta_t - \eta_{t-1}\|^2 + \frac{3\gamma^3 L^2 \beta_1^2}{(1-\beta_1)^2}\frac{dG_\infty^2}{\epsilon^3}. \tag{19}$$

Then, telescoping it from $t = 0$ to $t = T - 1$, and assuming $\gamma$ is a constant, it follows that

$$\frac{1}{T}\sum_{t=0}^{T-1}\mathbb{E}\|\nabla f(x_t)\|^2 \leq \frac{2G(f(x_0) - f^*)}{\gamma T} + \frac{8G\gamma L}{\epsilon}\frac{\sigma^2}{b\epsilon} + \frac{45GL^2\rho_t^2}{\epsilon} + \frac{2G^3}{(1-\beta_1)T}d\left(\frac{1}{\epsilon} - \frac{1}{G}\right)$$

$$+ \frac{8G\gamma L}{\epsilon}\frac{L\rho_t^2}{\epsilon} + \frac{2(4 + (\frac{\beta_1}{1-\beta_1})^2)\gamma LG^3}{T}d(\epsilon^{-2} - G^{-2}) + \frac{6\gamma^2 L^2 \beta_1^2}{(1-\beta_1)^2}\frac{dG^3}{\epsilon^3}, \tag{20}$$

which completes the proof.

## 4 EXPERIMENTS

In this section, we apply AdaSAM to training language models and compare it with SGD, AMSGrad, and SAM to show the effectiveness of AdaSAM. Due to space limitations, more experimental results visualization, task description, implementation details and ablation study are placed in the Appendix.

Table 1: Evaluating AMSGrad and AdaSAM on the GLUE benchmark with $\beta_1 = 0.9$

| Tasks | Metric | AMSGrad | AdaSAM rho=0.01 | AdaSAM rho=0.005 | AdaSAM rho=0.001 | AdaSAM best |
|---|---|---|---|---|---|---|
| CoLA | Mcc. | 68 | 65.29 | 68.74 | 67.3 | 68.74 |
| SST-2 | Acc. | 96.33 | 96.33 | 96.67 | 96.1 | 96.67 |
| MRPC | Acc. | 90.2 | 91.18 | 90.93 | 90.2 | 91.18 |
|  | F1 | 92.72 | 93.64 | 93.36 | 92.96 | 93.64 |
| STS-B | Pear.C. | 91.72 | 90.13 | 91.64 | 91.9 | 91.9 |
|  | Spea.C. | 91.48 | 90.36 | 91.38 | 91.62 | 91.62 |
| RTE | Acc. | 87.73 | 84.84 | 87.73 | 85.92 | 87.73 |
| MNLI-m | Acc. | 90.67 | 90.97 | 90.88 | 90.45 | 90.97 |
| MNLI-mm | Acc. | 90.41 | 90.42 | 90.4 | 90.4 | 90.42 |
| QNLI | Acc. | 94.82 | 94.65 | 94.56 | 94.56 | 94.65 |
| QQP | F1 | 88.7 | 88.55 | 88.69 | 88.64 | 88.69 |
|  | Acc. | 91.41 | 91.23 | 91.33 | 91.27 | 91.33 |
| Average Scores |  | 89.52 | 88.97 | 89.69 | 89.28 | 89.8 |
| relative improvements |  | – | -0.55 | 0.17 | -0.24 | 0.28 |

### 4.1 EXPERIMENTAL SETUP

**Tasks and Datasets.** We evaluate the SAM with an adaptive learning rate on a popular benchmark, *i.e.* General Language Understanding Evaluation (GLUE) Wang et al. (2018), which consists of several language understanding tasks including sentiment analysis, question answering and textual entailment. For a fair comparison, we report the results based on single-task, without multi-task or ensemble training. We evaluate the performance with Accuracy ("*Acc*") metric for most tasks, except the F1 scores for QQP and MRPC, the Pearson-Spearman correlations ("*Pcor/Scor*") for STS-B and the Matthew correlations ("*Mcc*") for CoLA.

Table 2: Evaluating SGD and SAM on the GLUE benchmark with $\beta_1 = 0.9$

| Tasks | Metric | AMSGrad | AdaSAM rho=0.01 | AdaSAM rho=0.005 | AdaSAM best |
|---|---|---|---|---|---|
| CoLA | Mcc. | 9.25 | 4.64 | 66.76 | 66.76 |
| SST-2 | Acc. | 50.92 | 95.87 | 95.76 | 95.87 |
| MRPC | Acc. | 68.38 | 70.58 | 68.38 | 70.58 |
| | F1 | 81.22 | 81.98 | 81.22 | 81.98 |
| STS-B | Pear.C. | 3.22 | 84.74 | 2 | 84.74 |
| | Spea.C. | 1.9 | 85.57 | 2 | 85.57 |
| RTE | Acc. | 55.6 | 52.71 | 52.71 | 52.71 |
| MNLI-m | Acc. | 84.94 | 90.5 | 90.42 | 90.5 |
| MNLI-mm | Acc. | 84.87 | 90.19 | 89.74 | 90.19 |
| QNLI | Acc. | 63.61 | 94.44 | 94.6 | 94.6 |
| QQP | F1 | 85.6 | 84.7 | 86.72 | 86.72 |
| | Acc. | 80.14 | 87.88 | 89.94 | 89.94 |
| Average Scores | | 55.8 | 76.98 | 68.35 | 82.51 |
| relative improvements | | – | 21.18 | 12.55 | 26.71 |

**Implementations.** We conduct our experiments using a widely-used pretrained language model, RoBERTa-large[1] in the open-source toolkit fairseq[2], with 24 transformer layers, a hidden size of 1024. For fine-tuning on each task, we use different combinations of hyper-parameters, including the learning rate, the number of epochs, the batch size, *etc* [3]. In particular, for RTE, STS-B and MRPC of GLUE benchmark, we first fine-tune the pre-trained RoBERTa-large model on the MNLI dataset and continue fine-tuning the RoBERTa-large-MNLI model on the corresponding single-task corpus for better performance, as many prior works did Liu et al. (2019); He et al. (2020). All models are trained on NVIDIA DGX SuperPOD cluster, in which each machine contains 8 x 40GB A100 GPUs.

Table 3: Evaluating AMSGrad and AdaSAM on the GLUE benchmark without momentum ($\beta_1 = 0$)

| Tasks | Metric | AMSGrad | AdaSAM(rho=0.01) | AdaSAM(rho=0.005) | AdaSAM(best) |
|---|---|---|---|---|---|
| CoLA | Mcc. | 63.78 | 69.23 | 68.47 | 69.23 |
| SST-2 | Acc. | 96.44 | 96.22 | 96.22 | 96.22 |
| MRPC | Acc. | 89.71 | 89.96 | 89.96 | 89.96 |
| | F1 | 92.44 | 92.84 | 92.82 | 92.84 |
| STS-B | Pear.C. | 89.98 | 88.83 | 91.59 | 91.59 |
| | Spea.C. | 90.35 | 89.07 | 91.22 | 91.22 |
| RTE | Acc. | 87.36 | 87 | 73.65 | 87 |
| MNLI-m | Acc. | 90.65 | 90.83 | 90.75 | 90.83 |
| MNLI-mm | Acc. | 90.35 | 90.41 | 90.42 | 90.42 |
| QNLI | Acc. | 94.53 | 94.8 | 94.73 | 94.8 |
| QQP | F1 | 88.59 | 88.67 | 88.72 | 88.72 |
| | Acc. | 91.27 | 91.38 | 91.46 | 91.46 |
| Average Scores | | 88.79 | 89.1 | 88.33 | 89.52 |
| relative improvements | | | 0.32 | -0.45 | 0.74 |

## 4.2 RESULTS ON GLUE BENCHMARK

We conduct the experiments on the SGD based method and adaptive learning rate method, respectively. Each method contains SAM and the base optimizer. Table 1 shows the AMSGrad and SAM with adaptive learning rate. For the SAM with an adaptive learning rate, we tune the neighborhood size from 0.01, 0.005, 0.001. The result shows that SAM with an adaptive learning rate outperforms AMSGrad on 6 tasks of 8 tasks except for QNLI and QQP. Overall, it improves the 0.28 than AMSGrad. Table 2 shows the result of SGD and SAM. We

---

[1] https://dl.fbaipublicfiles.com/fairseq/models/roberta.large.tar.gz
[2] https://github.com/facebookresearch/fairseq
[3] Due to the space limitation, we show the details of the dataset and training setting in Appendix A.

Table 4: Evaluating SGD and SAM on the GLUE benchmark $\beta_1 = 0$

| Tasks | Metric | SGD | SAM(rho=0.01) | SAM(rho=0.005) | SAM(best) |
|---|---|---|---|---|---|
| CoLA | Mcc. | 0 | 41.91 | 58.79 | 58.79 |
| SST-2 | Acc. | 51.72 | 95.3 | 81.54 | 95.3 |
| MRPC | Acc. | 68.38 | 68.38 | 68.38 | 68.38 |
| | F1 | 81.22 | 81.22 | 81.22 | 81.22 |
| STS-B | Pear.C. | 5.55 | 9.21 | 13.52 | 13.52 |
| | Spea.C. | 7.2 | 10.38 | 16.6 | 16.6 |
| RTE | Acc. | 51.27 | 53.07 | 53.79 | 53.79 |
| MNLI-m | Acc. | 32.51 | 87.99 | 88.42 | 88.42 |
| MNLI-mm | Acc. | 32.42 | 87.8 | 88.15 | 88.15 |
| QNLI | Acc. | 53.32 | 51.24 | 92.95 | 92.95 |
| QQP | F1 | 0 | 83.44 | 83.84 | 83.84 |
| | Acc. | 63.18 | 87.27 | 87.7 | 87.7 |
| Average Scores | | 37.23 | 63.1 | 67.91 | 69.06 |
| relative improvements | | - | 25.87 | 30.68 | 31.82 |

tune the neighborhood size from 0.01, 0.005 for SAM. The results show that SAM is better than SGD on 7 tasks of 8 tasks except for RTE. And SAM can significantly improve the performance.

Comparing the results of Table 1 and Table 2, we can find that the adaptive learning rate method is better than SGD based method. SAM with an adaptive learning rate achieves the best metric on 6 tasks. In general, SAM with an adaptive learning rate is better than the other methods.

In addition, we conduct the experiments the momentum is set to 0 to evaluate the influence of the adaptive learning rate. Table 3 shows that SAM with adaptive learning rate outperforms AMSGrad on 6 tasks of 8 tasks except for SST-2 and RTE. In Table 4, we compare SGD and SAM, and without the momentum, SAM outperforms SGD on all tasks. Under this situation, SAM with an adaptive learning rate method is better than the other methods.

When comparing the result of Table 1 and Table 3, adaptive learning rate method is more important than the momentum. When there is no momentum term, SAM with adaptive learning rate improves 0.74 than AMSGrad. With a momentum term, SAM with adaptive learning rate improves only 0.28.

### 4.3 MINI-BATCH SPEEDUP

In this part, we test the performance with different batch sizes to validate the linear speedup property. The experiments are conducted on the MRPC, RTE, and CoLA tasks. The batch size is set as 4, 8, 16, 32, respectively. We scale the learning rate as $\sqrt{N}$, which is similar as Li et al. (2021). $N$ is the batch size. The results show that the training loss decreases faster as the batchsize increases, and the loss curve with the batch size of 32 achieves nearly half iterations as the curve with the batch size of 16.

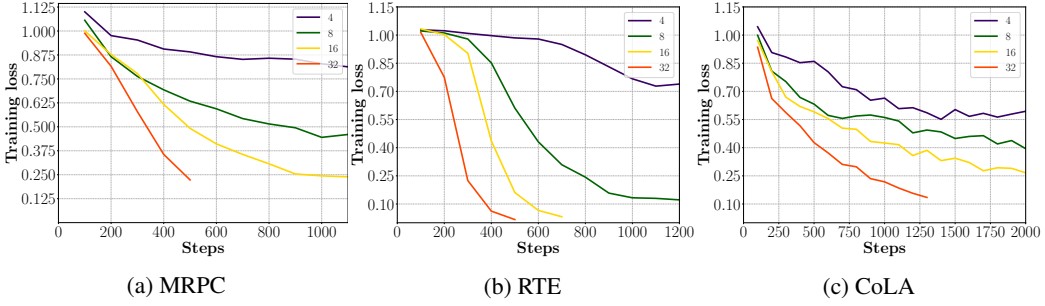

| (a) MRPC | (b) RTE | (c) CoLA |
|---|---|---|

Figure 1: The linear speedup of SAM-AMSGrad with the number of batch size of 4, 8, 16, 32.

## 5 CONCLUSION

In this work, we study the convergence rate of Sharpness aware minimization (SAM) optimizer with adaptive learning rate from AMSGrad in the stochastic non-convex setting. To the best of our knowledge, we are the first to provide the non-trivial $\mathcal{O}(1/\sqrt{bT})$ convergence rate of SAM with an adaptive learning rate, which achieves a linear speedup property with respect to mini-batch size $b$. We have conducted extensive experiments on several NLP tasks, which verifies that AdaSAM could achieve superior performance compared with AMSGrad and SAM optimizers. Future works include extending AdaSAM to the distributed setting and reducing the twice gradient back-propagation cost.

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

## A EXPERIMENTAL SETTINGS

Table 5: Experimental settings and data divisions upon different downstream tasks. Notably, for each tasks in GLUE benchmark, we provide the number of classes ("classes"), the learning rate ("lr"), the batch size ("bsz"), the total number of updates ("total"), the number of warmup updates ("warmup") and the number of GPUs ("GPUs") during fine-tuning, respectively.

| | MNLI | QNLI | QQP | RTE | SST-2 | MRPC | CoLA | STS-B |
|---|---|---|---|---|---|---|---|---|
| *experimental settings upon different downstream tasks* | | | | | | | | |
| –classes | 3 | 2 | 2 | 2 | 2 | 2 | 2 | 1 |
| –lr | 1e-5 | 1e-5 | 1e-5 | 2e-5 | 1e-5 | 1e-5 | 1e-5 | 2e-5 |
| –bsz | 256 | 128 | 256 | 32 | 64 | 32 | 32 | 32 |
| –total | 15,484 | 8,278 | 14,453 | 1,018 | 10,467 | 1,148 | 2,668 | 1,799 |
| –warmup | 929 | 496 | 867 | 61 | 628 | 68 | 160 | 107 |
| –GPUs | 4 | 4 | 8 | 2 | 2 | 2 | 2 | 2 |
| *data divisions for each dataset* | | | | | | | | |
| train | 392,720 | 104,743 | 363,870 | 2,491 | 67,350 | 5,801 | 8,551 | 5,749 |
| dev | 9,815 | 5,463 | 40,431 | 277 | 873 | 4,076 | 1,043 | 1,500 |
| test | 9,796 | 5,461 | 390,956 | 3,000 | 1,821 | 1,725 | 1,063 | 1,379 |

The GLUE benchmark contains 8 tasks, they are RTE, STS-B, CoLA, SST-2, MNLI, MRPC, QNLI and QQP. CoLA is a single sentence task. Each sentence has a label 1 and -1. 1 represents that it is a grammatical sentence, while -1 represents that it is illegal. Matthews correlation coefficient, dubbed **mcc** is used as our evaluation metric. STS-B is a similarity and paraphrase task. Each sample has a pair of a paragraph. People annotated the sample from 1 to 5 based on the similarity between the two paragraphs. The metric is Pearson and Spearman, dubbed **p/s** correlation coefficients. RTE is an inference task. Each sample has two sentences. If two sentences have a relation of entailment, we view them as a positive sample. If not, they compose of a negative sample. In the RTE task, the metric is the accuracy, dubbed **acc**. SST-2 is a single sentence task and its metric is the accuracy. MNLI is a sentence-level task that has 3 classes. They are entailment, contradiction and neutral. MRPC is a task to classify whether the sentences in the pair are equivalent. QNLI is a question-answering task. If the sentence contains the answer to the question, then it is a positive sample. QQP is a social question-answering task that consists of question pairs from Quora. It determines whether the questions are equivalent. The metric of MNLI, MRPC, QNLI, QQP is accuracy.

## B PROOF

We set $z_t = x_t + \frac{\beta_1}{1-\beta_1}(x_t - x_{t-1})$ for $t \geq 0$ and we assume $x_{-1} = 0$ and $m_{-1} = 0$.

We have that

$$z_{t+1} - z_t = x_{t+1} + \frac{\beta_1}{1-\beta_1}(x_{t+1} - x_t) - x_t - \frac{\beta_1}{1-\beta_1}(x_t - x_{t-1}) \tag{21}$$

$$= \frac{1}{1-\beta_1}(x_{t+1} - x_t) - \frac{\beta_1}{1-\beta_1}(x_t - x_{t-1}) \tag{22}$$

$$= -\frac{1}{1-\beta_1}\gamma m_t \odot \eta_t + \frac{\beta_1}{1-\beta_1}(x_t - x_{t-1})\gamma m_{t-1} \odot \eta_{t-1} \tag{23}$$

$$= -\frac{1}{1-\beta_1}\gamma(\beta_1 m_{t-1} + (1-\beta_1)g_t) \odot \eta_t + \frac{\beta_1}{1-\beta_1}(x_t - x_{t-1})\gamma m_{t-1} \odot \eta_{t-1} \tag{24}$$

$$= \frac{\beta_1}{1-\beta_1}\gamma m_{t-1} \odot (\eta_{t-1} - \eta_t) - \gamma g_t \odot \eta_t \tag{25}$$

By applying L-smooth, we have

$$f(z_{t+1}) \leq f(z_t) + \langle \nabla f(z_t), z_{t+1} - z_t \rangle + \frac{L}{2}\|z_{t+1} - z_t\|^2 \tag{26}$$

We re-organize it, and we have

$$f(z_{t+1}) - f(z_t)$$

$$\leq \langle \nabla f(z_t), z_{t+1} - z_t \rangle + \frac{L}{2}\|z_{t+1} - z_t\|^2 \tag{27}$$

$$= \langle \nabla f(z_t), \frac{\gamma \beta_1}{1-\beta_1} m_{t-1} \odot (\eta_{t-1} - \eta_t) \rangle + \langle \nabla f(z_t), -\gamma g_t \odot \eta_t \rangle + \frac{L}{2} \|z_{t+1} - z_t\|^2 \tag{28}$$

$$\begin{aligned} = {} & \langle \nabla f(z_t), \frac{\gamma \beta_1}{1-\beta_1} m_{t-1} \odot (\eta_{t-1} - \eta_t) \rangle + \frac{L}{2} \|z_{t+1} - z_t\|^2 \\ & + \langle \nabla f(z_t), \frac{\gamma_t}{b} \sum_{i \in B} \nabla f_i(x_t + \rho_t \frac{s_t}{\|s_t\|}) \odot (\eta_{t-1} - \eta_t) \rangle \\ & + \langle \nabla f(z_t), -\frac{\gamma_t}{b} \sum_{i \in B} \nabla f_i(x_t + \rho_t \frac{s_t}{\|s_t\|}) \odot \eta_{t-1} \rangle \end{aligned} \tag{29}$$

$$\begin{aligned} = {} & \langle \nabla f(z_t), \frac{\gamma \beta_1}{1-\beta_1} m_{t-1} \odot (\eta_{t-1} - \eta_t) \rangle + \frac{L}{2} \|z_{t+1} - z_t\|^2 \\ & + \langle \nabla f(z_t), \frac{\gamma_t}{b} \sum_{i \in B} \nabla f_i(x_t + \rho_t \frac{s_t}{\|s_t\|}) \odot (\eta_{t-1} - \eta_t) \rangle \\ & + \langle \nabla f(z_t) - \nabla f(x_t), -\frac{\gamma_t}{b} \sum_{i \in B} \nabla f_i(x_t + \rho_t \frac{s_t}{\|s_t\|}) \odot \eta_{t-1} \rangle \\ & + \langle \nabla f(x_t), -\frac{\gamma_t}{b} \sum_{i \in B} \nabla f_i(x_t + \rho_t \frac{s_t}{\|s_t\|}) \odot \eta_{t-1} \rangle \end{aligned} \tag{30}$$

$$\begin{aligned} = {} & \langle \nabla f(z_t), \frac{\gamma \beta_1}{1-\beta_1} m_{t-1} \odot (\eta_{t-1} - \eta_t) \rangle + \frac{L}{2} \|z_{t+1} - z_t\|^2 \\ & + \langle \nabla f(z_t), \frac{\gamma_t}{b} \sum_{i \in B} \nabla f_i(x_t + \rho_t \frac{s_t}{\|s_t\|}) \odot (\eta_{t-1} - \eta_t) \rangle \\ & + \langle \nabla f(z_t) - \nabla f(x_t), -\frac{\gamma_t}{b} \sum_{i \in B} \nabla f_i(x_t + \rho_t \frac{\sum \nabla f_i(x_t)}{\|\sum \nabla f_i(x_t)\|}) \odot \eta_{t-1} \rangle \\ & + \langle \nabla f(x_t), \frac{\gamma_t}{b} \sum_{i \in B} \nabla f_i(x_t + \rho_t \frac{\nabla f(x_t)}{\|\nabla f(x_t)\|}) \odot \eta_{t-1} - \frac{\gamma_t}{b} \sum_{i \in B} \nabla f_i(x_t + \rho_t \frac{s_t}{\|s_t\|}) \odot \eta_{t-1} \rangle \\ & + \langle \nabla f(x_t), -\frac{\gamma_t}{b} \sum_{i \in B} \nabla f_i(x_t + \rho_t \frac{\nabla f(x_t)}{\|\nabla f(x_t)\|}) \odot \eta_{t-1} \rangle. \end{aligned} \tag{31}$$

From the Lemma B.5, Lemma B.6, Lemma B.7, we have

$$\langle \nabla f(z_t), \frac{\gamma_t}{b} \sum_{i \in B} \nabla f_i(x_t + \rho_t \frac{s_t}{\|s_t\|}) \odot (\eta_{t-1} - \eta_t) \rangle \leq \gamma_t G^2 \|\eta_{t-1} - \eta_t\|_1, \tag{32}$$

$$\langle \nabla f(z_t), \frac{\gamma \beta_1}{1-\beta_1} m_{t-1} \odot (\eta_{t-1} - \eta_t) \rangle \leq \frac{\gamma \beta_1}{1-\beta_1} G^2 \|\eta_{t-1} - \eta_t\|_1, \tag{33}$$

$$\langle \nabla f(x_t), \frac{\eta_t}{b} \sum_{i \in B} \nabla f_i(x_t + \rho_t \frac{\nabla f(x_t)}{\|\nabla f(x_t)\|}) \odot \eta_{t-1} - \frac{\gamma_t}{b} \sum_{i \in B} \nabla f_i(x_t + \rho_t \frac{s_t}{\|s_t\|}) \odot \eta_{t-1} \rangle$$
$$\leq \frac{\gamma_t}{2\mu^2} \|\nabla f(x_t) \odot \sqrt{\eta_{t-1}}\|^2 + \frac{2\mu^2 \gamma_t L^2 \rho_t^2}{\epsilon}. \tag{34}$$

Taking conditional expectation, we have

$$\mathbb{E} f(z_{t+1}) - f(z_t) \tag{35}$$
$$\leq \mathbb{E} \langle \nabla f(x_t), -\frac{\gamma_t}{b} \sum_{i \in B} \nabla f_i(x_t + \rho_t \frac{\nabla f(x_t)}{\|\nabla f(x_t)\|}) \odot \eta_{t-1} \rangle + \frac{L}{2} \mathbb{E} \|z_{t+1} - z_t\|^2$$
$$+ \frac{\gamma_t}{2\mu^2} \|\nabla f(x_t) \odot \sqrt{\eta_{t-1}}\|^2 + \frac{2\mu^2 \gamma_t L^2 \rho_t^2}{\epsilon} + \frac{\gamma}{1-\beta_1} G^2 \|\eta_{t-1} - \eta_t\|_1$$
$$+ \mathbb{E} \langle \nabla f(z_t) - \nabla f(x_t), -\frac{\gamma_t}{b} \sum_{i \in B} \nabla f_i(x_t + \rho_t \frac{s_t}{\|s_t\|}) \odot \eta_{t-1} \rangle \tag{36}$$

where $\mu > 0$ is to be determined.

For the term

$$\mathbb{E} \langle \nabla f(x_t), -\frac{\gamma_t}{b} \sum_{i \in B} \nabla f_i(x_t + \rho_t \frac{\nabla f(x_t)}{\|\nabla f(x_t)\|}) \odot \eta_{t-1} \rangle, \tag{37}$$

the term

$$\frac{L}{2}\mathbb{E}\|z_{t+1} - z_t\|^2, \tag{38}$$

and the term

$$\mathbb{E}\langle \nabla f(z_t) - \nabla f(x_t), -\frac{\gamma_t}{b}\sum_{i\in B}\nabla f_i(x_t + \rho_t \frac{s_t}{\|s_t\|}) \odot \eta_{t-1}\rangle, \tag{39}$$

we introduce the Lemma B.8, the Lemma B.10 and the Lemma B.9. We take the expectation over the whole processing and we have

$$\mathbb{E}f(z_{t+1}) - \mathbb{E}f(z_t)$$

$$\leq \frac{\gamma_t}{2\mu^2}\mathbb{E}\|\nabla f(x_t) \odot \sqrt{\eta_{t-1}}\|^2 + \frac{2\mu^2\gamma_t L^2\rho_t^2}{\epsilon} + \frac{\gamma}{1-\beta_1}G^2\mathbb{E}\|\eta_{t-1} - \eta_t\|_1$$

$$- \gamma_t\mathbb{E}\|\nabla f(x_t) \odot \sqrt{\eta_{t-1}}\|^2 + \mathbb{E}\frac{\gamma_t}{2\alpha^2}\mathbb{E}\|\nabla f(x_t) \odot \sqrt{\eta_{t-1}}\|^2 + \frac{\gamma_t\alpha^2 L^2\rho^2}{2\epsilon} + \frac{LG^2\gamma^2\beta_1^2}{(1-\beta_1)^2}\mathbb{E}\|\eta_t - \eta_{t-1}\|^2$$

$$+ \gamma_t^2 L(3\frac{1+\beta}{\beta\epsilon}(\mathbb{E}\|\nabla f(x_t) \odot \sqrt{\eta_{t-1}}\|^2 + \frac{L\rho_t^2}{\epsilon} + \frac{\sigma^2}{b\epsilon}) + (1+\beta)G^2\mathbb{E}\|\eta_t - \eta_{t-1}\|^2)$$

$$+ \frac{\gamma^3 L^2\beta_1^2}{2\epsilon(1-\beta_1)^2}(\frac{1}{\lambda_1^2} + \frac{1}{\lambda_2^2} + \frac{1}{\lambda_3^2})\frac{dG_\infty^2}{\epsilon^2} + \frac{\gamma\lambda_1^2}{2}\|\nabla f(x_t) \odot \sqrt{\eta_{t-1}}\|^2 + \frac{\gamma L^2\rho_t^2}{2\epsilon}(\lambda_2^2 + 4\lambda_3^2) \tag{40}$$

$$= -\gamma_t(1 - \frac{1}{2\mu^2} - \frac{1}{2\alpha^2} - \frac{3\gamma L(1+\beta)}{\beta\epsilon} - \frac{\lambda_1^2}{2})\mathbb{E}\|\nabla f(x_t) \odot \sqrt{\eta_{t-1}}\|^2 + \frac{2\mu^2\gamma_t L^2\rho_t^2}{\epsilon} + \frac{\gamma}{1-\beta_1}G^2\mathbb{E}\|\eta_{t-1} - \eta_t\|_1$$

$$+ \frac{\gamma_t\alpha^2 L^2\rho^2}{2\epsilon} + \frac{3\gamma_t^2 L(1+\beta)}{\beta\epsilon}(\frac{L\rho_t^2}{\epsilon} + \frac{\sigma^2}{b\epsilon}) + \gamma_t^2 LG^2((\frac{\beta_1}{1-\beta_1})^2 + 1 + \beta)\mathbb{E}\|\eta_t - \eta_{t-1}\|^2$$

$$+ \frac{\gamma^3 L^2\beta_1^2}{2\epsilon(1-\beta_1)^2}(\frac{1}{\lambda_1^2} + \frac{1}{\lambda_2^2} + \frac{1}{\lambda_3^2})\frac{dG_\infty^2}{\epsilon^2} + \frac{\gamma L^2\rho_t^2}{2\epsilon}(\lambda_2^2 + 4\lambda_3^2). \tag{41}$$

We set $\mu^2 = \alpha^2 = 8$, $\beta = 3$, $\lambda_1^2 = \frac{1}{4}$, $\lambda_2^2 = \lambda_3^2 = 1$ and we choose $\frac{2\gamma_t L}{\epsilon} \leq \frac{1}{8}$. So we have

$$\mathbb{E}f(x_{t+1}) - \mathbb{E}f(x_t)$$

$$\leq -\frac{\gamma_t}{2}\mathbb{E}\|\nabla f(x_t) \odot \sqrt{\eta_{t-1}}\|^2 + \frac{16\gamma_t L^2\rho_t^2}{\epsilon} + \frac{\gamma}{1-\beta_1}G^2\mathbb{E}\|\eta_{t-1} - \eta_t\|_1$$

$$+ \frac{4\gamma_t L^2\rho^2}{\epsilon} + \frac{4\gamma_t^2 L}{\epsilon}(\frac{L\rho_t^2}{\epsilon} + \frac{\sigma^2}{b\epsilon}) + (4 + (\frac{\beta_1}{1-\beta_1})^2)\gamma_t^2 LG^2\mathbb{E}\|\eta_t - \eta_{t-1}\|^2$$

$$+ \frac{3\gamma^3 L^2\beta_1^2}{\epsilon(1-\beta_1)^2}\frac{dG_\infty^2}{\epsilon^2} + \frac{5\gamma L^2\rho_t^2}{2\epsilon} \tag{42}$$

We re-arrange it and $\eta_t$ is bounded. We have

$$\frac{\gamma_t}{2G}\mathbb{E}\|\nabla f(x_t)\|^2 \leq \frac{\gamma_t}{2}\mathbb{E}\|\nabla f(x_t) \odot \sqrt{\eta_{t-1}}\|^2 \tag{43}$$

$$\leq -\mathbb{E}f(x_{t+1}) + \mathbb{E}f(x_t) + \frac{45\gamma_t L^2\rho_t^2}{2\epsilon} + \frac{\gamma}{1-\beta_1}G^2\mathbb{E}\|\eta_{t-1} - \eta_t\|_1$$

$$+ \frac{4\gamma_t^2 L}{\epsilon}(\frac{L\rho_t^2}{\epsilon} + \frac{\sigma^2}{b\epsilon}) + (4 + (\frac{\beta_1}{1-\beta_1})^2)\gamma_t^2 LG^2\mathbb{E}\|\eta_t - \eta_{t-1}\|^2 + \frac{3\gamma^3 L^2\beta_1^2}{(1-\beta_1)^2}\frac{dG_\infty^2}{\epsilon^3}. \tag{44}$$

We summary it from $t = 0$ to $t = T - 1$, and we assume $\gamma_t$ is a constant, and we have

$$\frac{1}{T}\sum_{t=0}^{T-1}\mathbb{E}\|\nabla f(x_t)\|^2 \leq 2G\frac{\mathbb{E}f(x_0) - \mathbb{E}f(x_{t+1})}{\gamma_t T} + \frac{45GL^2\rho_t^2}{\epsilon} + \frac{2G^3}{(1-\beta_1)T}\mathbb{E}\sum_{t=0}^{T-1}\|\eta_{t-1} - \eta_t\|_1$$

$$+ \frac{8G\gamma_t L}{\epsilon}(\frac{L\rho_t^2}{\epsilon} + \frac{\sigma^2}{b\epsilon}) + \frac{2(4 + (\frac{\beta_1}{1-\beta_1})^2)\gamma_t LG^3}{T}\mathbb{E}\sum_{t=0}^{T-1}\|\eta_t - \eta_{t-1}\|^2 + \frac{6\gamma^2 L^2\beta_1^2}{(1-\beta_1)^2}\frac{dG^3}{\epsilon^3} \tag{45}$$

$$\leq \frac{2G(f(x_0) - f^*)}{\gamma_t T} + \frac{45GL^2\rho_t^2}{\epsilon} + \frac{2G^3}{(1-\beta_1)T}d(\frac{1}{\epsilon} - \frac{1}{G}) + \frac{8G\gamma_t L}{\epsilon}(\frac{L\rho_t^2}{\epsilon} + \frac{\sigma^2}{b\epsilon})$$

$$+ \frac{2(4 + (\frac{\beta_1}{1-\beta_1})^2)\gamma_t LG^3}{T}d(\epsilon^{-2} - G^{-2}) + \frac{6\gamma^2 L^2\beta_1^2}{(1-\beta_1)^2}\frac{dG^3}{\epsilon^3} \tag{46}$$

$$= \frac{2G(f(x_0) - f^*)}{\gamma_t T} + \frac{8G\gamma_t L}{\epsilon}\frac{\sigma^2}{b\epsilon} + \frac{45GL^2\rho_t^2}{\epsilon} + \frac{2G^3}{(1-\beta_1)T}d(\frac{1}{\epsilon} - \frac{1}{G}) + \frac{8G\gamma_t L}{\epsilon}\frac{L\rho_t^2}{\epsilon}$$

$$+ \frac{2(4 + (\frac{\beta_1}{1-\beta_1})^2)\gamma_t LG^3}{T}d(\epsilon^{-2} - G^{-2}) + \frac{6\gamma^2 L^2\beta_1^2}{(1-\beta_1)^2}\frac{dG^3}{\epsilon^3}. \tag{47}$$

## B.1 TECHNICAL LEMMA

**Lemma B.1.** *Given two vectors $a$, $b \in \mathbb{R}^d$, we have $\langle a, b \rangle \leq \frac{\lambda^2}{2}\|a\|^2 + \frac{1}{2\lambda^2}\|b\|^2$ for parameter $\lambda$, $\forall \lambda \in (1, +\infty)$.*

*Proof.*

$$RHS = \frac{\lambda^2}{2}\sum_{j=1}^{d}(a)_j^2 + \frac{1}{2\lambda^2}\sum_{j=1}^{d}(b)_j^2 \geq \sum_{j=1}^{d} 2\sqrt{\frac{\lambda^2}{2}(a)_j^2 \times \frac{1}{2\lambda^2}(b)_j^2} = \sum_{j=1}^{d}|(a)_j| \times |(b)_j| \geq LHS. \quad (48)$$

□

**Lemma B.2.** *For any vector $x, y \in \mathbb{R}^d$, we have*

$$\|x \odot y\|^2 \leq \|x\|^2 \times \|y\|_\infty^2 \leq \|x\|^2 \times \|y\|^2. \quad (49)$$

*Proof.* The first inequality can be derived from that $\sum_{i=1}^{d}(x_i^2 y_i^2) \leq \sum_{i=1}^{d}(x_i^2\|y\|_\infty^2)$. The second inequality follows from that $\|y\|_\infty^2 \leq \|y\|^2$. □

**Lemma B.3.** *$\eta$ is bounded, i.e., $\frac{1}{G_\infty} \leq (\eta_t)_j \leq \frac{1}{\epsilon}$.*

*Proof.* As the gradient is bounded by $G$ and $(\eta_t)_j = \frac{1}{\sqrt{(\hat{v}_t)_j}}$. Follow the update rule, we have $\frac{1}{G_\infty} \leq (\eta_t)_j \leq \frac{1}{\epsilon}$. □

**Lemma B.4.** *For the term defined in the algorithm, we have*

$$\frac{1}{T}\mathbb{E}\sum_{t=0}^{T-1}\|\eta_{t-1} - \eta_t\|^1 \leq \frac{d}{T}\left(\frac{1}{\epsilon} - \frac{1}{G}\right) \quad (50)$$

*Proof.* $(\eta_t)_i$, the i-th dimension of $\eta_t$ deceases as t increases. So we have

$$\frac{1}{T}\mathbb{E}\sum_{t=0}^{T-1}\|\eta_{t-1} - \eta_t\|^1 = \mathbb{E}\frac{1}{T}\sum_{i=1}^{d}\sum_{t=0}^{T-1}|(\eta_{t-1})_i - (\eta_t)_i|$$

$$\leq \mathbb{E}\frac{1}{T}\sum_{i=1}^{d}((\eta_{-1})_i - (\eta_{T-1})_i) \leq \mathbb{E}\frac{1}{T}\sum_{i=1}^{d}\left(\frac{1}{\epsilon} - \frac{1}{G}\right) = \frac{d}{T}\left(\frac{1}{\epsilon} - \frac{1}{G}\right) \quad (51)$$

□

**Lemma B.5.** *For the term defined in the algorithm, we have*

$$\langle \nabla f(z_t), \frac{\gamma_t}{b}\sum_{i \in B}\nabla f_i(x_t + \rho_t\frac{s_t}{\|s_t\|}) \odot (\eta_{t-1} - \eta_t)\rangle \leq \gamma_t G^2\|\eta_{t-1} - \eta_t\|_1 \quad (52)$$

*Proof.*

$$\langle \nabla f(z_t), \frac{\gamma_t}{b}\sum_{i \in B}\nabla f_i(x_t + \rho_t\frac{s_t}{\|s_t\|}) \odot (\eta_{t-1} - \eta_t)\rangle$$

$$\leq \gamma_t\sum_{j=1}^{d}|(\nabla f(z_t))_{(j)}| \times |(\frac{1}{b}\sum_{i \in B}\nabla f_i(x_t + \rho_t\frac{\sum \nabla f_i(x_t)}{\|\sum \nabla f_i(x_t)\|}) \odot (\eta_{t-1} - \eta_t))_{(j)}| \quad (53)$$

$$\leq \gamma_t G\sum_{j=1}^{d}|((\frac{1}{b}\sum_{i \in B}\nabla f_i(x_t + \rho_t\frac{\sum \nabla f_i(x_t)}{\|\sum \nabla f_i(x_t)\|}) \odot (\eta_{t-1} - \eta_t))_{(j)}| \quad (54)$$

$$\leq \frac{\gamma_t G}{b}\sum_{j=1}^{d}\sum_{i \in B}|((\nabla f_i(x_t + \rho_t\frac{\sum \nabla f_i(x_t)}{\|\sum \nabla f_i(x_t)\|}) \odot (\eta_{t-1} - \eta_t))_{(j)}| \quad (55)$$

$$= \frac{\gamma_t G}{b}\sum_{j=1}^{d}\sum_{i \in B}|(\nabla f_i(x_t + \rho_t\frac{\sum \nabla f_i(x_t)}{\|\sum \nabla f_i(x_t)\|})_{(j)} \times (\eta_{t-1} - \eta_t)_{(j)}| \quad (56)$$

$$\leq \frac{\gamma_t G^2}{b} \sum_{j=1}^{d} \sum_{i \in B} |(\eta_{t-1} - \eta_t)_{(j)}| \tag{57}$$

$$= \gamma_t G^2 \|\eta_{t-1} - \eta_t\|_1 \tag{58}$$

$\square$

**Lemma B.6.** *For the term defined in the algorithm, we have*

$$\langle \nabla f(z_t), \frac{\gamma \beta_1}{1 - \beta_1} m_{t-1} \odot (\eta_{t-1} - \eta_t) \rangle \leq \frac{\gamma \beta_1}{1 - \beta_1} G^2 \|\eta_{t-1} - \eta_t\|_1 \tag{59}$$

*Proof.*

$$\langle \nabla f(z_t), \frac{\gamma \beta_1}{1 - \beta_1} m_{t-1} \odot (\eta_{t-1} - \eta_t) \rangle$$

$$\leq \frac{\gamma \beta_1}{1 - \beta_1} \sum_{j=1}^{d} |(\nabla f(z_t))_{(j)}| \times |(m_{t-1} \odot (\eta_{t-1} - \eta_t))_{(j)}| \tag{60}$$

$$\leq \frac{\gamma \beta_1}{1 - \beta_1} G \sum_{j=1}^{d} |(m_{t-1} \odot (\eta_{t-1} - \eta_t))_{(j)}| \tag{61}$$

$$= \frac{\gamma \beta_1}{1 - \beta_1} \sum_{j=1}^{d} |(m_{t-1})_{(j)} \times (\eta_{t-1} - \eta_t)_{(j)}| \tag{62}$$

$$\leq \frac{\gamma \beta_1}{1 - \beta_1} G^2 \sum_{j=1}^{d} |(\eta_{t-1} - \eta_t)_{(j)}| \tag{63}$$

$$= \frac{\gamma \beta_1}{1 - \beta_1} G^2 \|\eta_{t-1} - \eta_t\|_1 \tag{64}$$

$\square$

**Lemma B.7.** *For the term defined in the algorithm, we have*

$$\langle \nabla f(x_t), \frac{\gamma_t}{b} \sum_{i \in B} \nabla f_i(x_t + \rho_t \frac{\nabla f(x_t)}{\|\nabla f(x_t)\|}) \odot \eta_{t-1} - \frac{\gamma_t}{b} \sum_{i \in B} \nabla f_i(x_t + \rho_t \frac{s_t}{\|s_t\|}) \odot \eta_{t-1} \rangle$$

$$\leq \frac{\gamma_t}{2\mu^2} \|\nabla f(x_t) \odot \sqrt{\eta_{t-1}}\|^2 + \frac{2\mu^2 \gamma_t L^2 \rho_t^2}{\epsilon}. \tag{65}$$

*Proof.*

$$\langle \nabla f(x_t), \frac{\gamma_t}{b} \sum_{i \in B} \nabla f_i(x_t + \rho_t \frac{\nabla f(x_t)}{\|\nabla f(x_t)\|}) \odot \eta_{t-1} - \frac{\gamma_t}{b} \sum_{i \in B} \nabla f_i(x_t + \rho_t \frac{s_t}{\|s_t\|}) \odot \eta_{t-1} \rangle$$

$$= \langle \nabla f(x_t) \odot \sqrt{\eta_{t-1}}, \frac{\gamma_t}{b} \sum_{i \in B} (\nabla f_i(x_t + \rho_t \frac{\nabla f(x_t)}{\|\nabla f(x_t)\|}) - \nabla f_i(x_t + \rho_t \frac{\sum_{i \in B} \nabla f_i(x_t)}{\|\sum_{i \in B} \nabla f_i(x_t)\|})) \odot \sqrt{\eta_{t-1}} \rangle \tag{66}$$

$$\leq \frac{\mu^2 \gamma_t}{2b^2} \|\sum (\nabla f_i(x_t + \rho_t \frac{\nabla f(x_t)}{\|\nabla f(x_t)\|}) - \nabla f_i(x_t + \rho_t \frac{\sum_{i \in B} \nabla f_i(x_t)}{\|\sum_{i \in B} \nabla f_i(x_t)\|})) \odot \sqrt{\eta_{t-1}}\|^2$$
$$+ \frac{\gamma_t}{2\mu^2} \|\nabla f(x_t) \odot \sqrt{\eta_{t-1}}\|^2 \tag{67}$$

$$\leq + \frac{\mu^2 \gamma_t}{2b} \sum \|\nabla f_i(x_t + \rho_t \frac{\nabla f(x_t)}{\|\nabla f(x_t)\|}) - \nabla f_i(x_t + \rho_t \frac{\sum_{i \in B} \nabla f_i(x_t)}{\|\sum_{i \in B} \nabla f_i(x_t)\|}) \odot \sqrt{\eta_{t-1}}\|^2$$
$$+ \frac{\gamma_t}{2\mu^2} \|\nabla f(x_t) \odot \sqrt{\eta_{t-1}}\|^2 \tag{68}$$

$$\leq + \frac{\mu^2 \gamma_t}{2b} \sum \|\nabla f_i(x_t + \rho_t \frac{\nabla f(x_t)}{\|\nabla f(x_t)\|}) - \nabla f_i(x_t + \rho_t \frac{\sum_{i \in B} \nabla f_i(x_t)}{\|\sum_{i \in B} \nabla f_i(x_t)\|})\|^2 \times \|\sqrt{\eta_{t-1}}\|_\infty^2$$
$$+ \frac{\gamma_t}{2\mu^2} \|\nabla f(x_t) \odot \sqrt{\eta_{t-1}}\|^2 \tag{69}$$

$$\leq \frac{\gamma_t}{2\mu^2}\|\nabla f(x_t) \odot \sqrt{\eta_{t-1}}\|^2 + \frac{\mu^2 \gamma_t L^2 \rho_t^2}{2b\epsilon}\sum\|\frac{\nabla f(x_t)}{\|\nabla f(x_t)\|} - \frac{\sum_{i \in B}\nabla f_i(x_t)}{\|\sum_{i \in B}\nabla f_i(x_t)\|}\|^2 \tag{70}$$

$$\leq \frac{\gamma_t}{2\mu^2}\|\nabla f(x_t) \odot \sqrt{\eta_{t-1}}\|^2 + \frac{2\mu^2 \gamma_t L^2 \rho_t^2}{\epsilon}. \tag{71}$$

$\square$

**Lemma B.8.** *For the term defined in the algorithm, we have*

$$\mathbb{E}\langle\nabla f(x_t), -\frac{\gamma_t}{b}\sum_{i \in B}\nabla f_i(x_t + \rho_t\frac{\nabla f(x_t)}{\|\nabla f(x_t)\|}) \odot \eta_{t-1}\rangle$$

$$\leq -\gamma_t\|\nabla f(x_t) \odot \sqrt{\eta_{t-1}}\|^2 + \mathbb{E}\frac{\gamma_t}{2\alpha^2}\|\nabla f(x_t) \odot \sqrt{\eta_{t-1}}\|^2 + \frac{\gamma_t \alpha^2 L^2 \rho_t^2}{2\epsilon} \tag{72}$$

*Proof.*

$$\mathbb{E}\langle\nabla f(x_t), -\frac{\gamma_t}{b}\sum_{i \in B}\nabla f_i(x_t + \rho_t\frac{\nabla f(x_t)}{\|\nabla f(x_t)\|}) \odot \eta_{t-1}\rangle$$

$$= -\gamma_t\|\nabla f(x_t) \odot \sqrt{\eta_{t-1}}\|^2 + \mathbb{E}\langle\nabla f(x_t), \frac{\gamma_t}{b}\sum_{i \in B}(\nabla f(x_t) - \nabla f_i(x_t + \rho_t\frac{\nabla f(x_t)}{\|\nabla f(x_t)\|})) \odot \eta_{t-1}\rangle \tag{73}$$

$$= -\gamma_t\|\nabla f(x_t) \odot \sqrt{\eta_{t-1}}\|^2 + \mathbb{E}\langle\nabla f(x_t), \frac{\gamma_t}{b}\sum_{i \in B}(\nabla f_i(x_t) - \nabla f_i(x_t + \rho_t\frac{\nabla f(x_t)}{\|\nabla f(x_t)\|})) \odot \eta_{t-1}\rangle \tag{74}$$

$$\leq -\gamma_t\|\nabla f(x_t) \odot \sqrt{\eta_{t-1}}\|^2 + \mathbb{E}\frac{\gamma_t}{2\alpha^2}\|\nabla f(x_t) \odot \sqrt{\eta_{t-1}}\|^2$$
$$+ \frac{\gamma_t \alpha^2}{2}\mathbb{E}\|\frac{1}{b}\sum_{i \in B}(\nabla f_i(x_t) - \nabla f_i(x_t + \rho_t\frac{\nabla f(x_t)}{\|\nabla f(x_t)\|})) \odot \sqrt{\eta_{t-1}}\|^2 \tag{75}$$

$$\leq -\gamma_t\|\nabla f(x_t) \odot \sqrt{\eta_{t-1}}\|^2 + \mathbb{E}\frac{\gamma_t}{2\alpha^2}\|\nabla f(x_t) \odot \sqrt{\eta_{t-1}}\|^2$$
$$+ \frac{\gamma_t \alpha^2}{2\epsilon}\mathbb{E}\|\frac{1}{b}\sum_{i \in B}(\nabla f_i(x_t) - \nabla f_i(x_t + \rho_t\frac{\nabla f(x_t)}{\|\nabla f(x_t)\|}))\|^2 \tag{76}$$

$$\leq -\gamma_t\|\nabla f(x_t) \odot \sqrt{\eta_{t-1}}\|^2 + \mathbb{E}\frac{\gamma_t}{2\alpha^2}\|\nabla f(x_t) \odot \sqrt{\eta_{t-1}}\|^2$$
$$+ \frac{\gamma_t \alpha^2}{2b\epsilon}\mathbb{E}\sum_{i \in B}\|(\nabla f_i(x_t) - \nabla f_i(x_t + \rho_t\frac{\nabla f(x_t)}{\|\nabla f(x_t)\|}))\|^2 \tag{77}$$

$$\leq -\gamma_t\|\nabla f(x_t) \odot \sqrt{\eta_{t-1}}\|^2 + \mathbb{E}\frac{\gamma_t}{2\alpha^2}\|\nabla f(x_t) \odot \sqrt{\eta_{t-1}}\|^2 + \frac{\gamma_t \alpha^2 L^2 \rho_t^2}{2b\epsilon}\mathbb{E}\sum_{i \in B}\|\frac{\nabla f(x_t)}{\|\nabla f(x_t)\|}\|^2 \tag{78}$$

$$= -\gamma_t\|\nabla f(x_t) \odot \sqrt{\eta_{t-1}}\|^2 + \mathbb{E}\frac{\gamma_t}{2\alpha^2}\|\nabla f(x_t) \odot \sqrt{\eta_{t-1}}\|^2 + \frac{\gamma_t \alpha^2 L^2 \rho_t^2}{2\epsilon} \tag{79}$$

$\square$

**Lemma B.9.** *For the term defined in the algorithm, we have*

$$\mathbb{E}\langle\nabla f(z_t) - \nabla f(x_t), -\frac{\gamma_t}{b}\sum_{i \in B}\nabla f_i(x_t + \rho_t\frac{s_t}{\|s_t\|}) \odot \eta_{t-1}\rangle$$

$$\leq \frac{\gamma^3 L^2 \beta_1^2}{2\epsilon(1-\beta_1)^2}(\frac{1}{\lambda_1^2} + \frac{1}{\lambda_2^2} + \frac{1}{\lambda_3^2})\frac{dG_\infty^2}{\epsilon^2} + \frac{\gamma\lambda_1^2}{2}\|\nabla f(x_t) \odot \sqrt{\eta_{t-1}}\|^2 + \frac{\gamma L^2 \rho_t^2}{2\epsilon}(\lambda_2^2 + 4\lambda_3^2). \tag{80}$$

*Proof.*

$$\mathbb{E}\langle\nabla f(z_t) - \nabla f(x_t), -\frac{\gamma_t}{b}\sum_{i \in B}\nabla f_i(x_t + \rho_t\frac{s_t}{\|s_t\|}) \odot \eta_{t-1}\rangle \tag{81}$$

$$= \gamma\mathbb{E}\langle(\nabla f(x_t) - \nabla f(z_t)) \odot \sqrt{\eta_{t-1}}, \frac{1}{b}\sum_{i \in B}\nabla f_i(x_t + \rho_t\frac{\sum_{i \in B}\nabla f_i(x_t)}{\|\sum_{i \in B}\nabla f_i(x_t)\|}) \odot \sqrt{\eta_{t-1}}\rangle \tag{82}$$

$$= \gamma\mathbb{E}\langle(\nabla f(x_t) - \nabla f(z_t)) \odot \sqrt{\eta_{t-1}}, \nabla f(x_t) \odot \sqrt{\eta_{t-1}}\rangle$$
$$+ \gamma\mathbb{E}\langle(\nabla f(x_t) - \nabla f(z_t)) \odot \sqrt{\eta_{t-1}}, \frac{1}{b}\sum_{i \in B}(\nabla f_i(x_t + \rho_t\frac{\nabla f(x_t)}{\|\nabla f(x_t)\|}) - \nabla f_i(x_t)) \odot \sqrt{\eta_{t-1}}\rangle$$

$$+ \gamma \mathbb{E} \langle (\nabla f(x_t) - \nabla f(z_t)) \odot \sqrt{\eta_{t-1}}, \frac{1}{b} \sum_{i \in B} (\nabla f_i(x_t + \rho_t \frac{\sum_{i \in B} \nabla f_i(x_t)}{\| \sum_{i \in B} \nabla f_i(x_t) \|}) - \nabla f_i(x_t + \rho_t \frac{\nabla f(x_t)}{\| \nabla f(x_t) \|}) \odot \sqrt{\eta_{t-1}} \rangle \tag{83}$$

$$\leq \frac{\gamma}{2}(\frac{1}{\lambda_1^2} + \frac{1}{\lambda_2^2} + \frac{1}{\lambda_3^2}) \mathbb{E} \| (\nabla f(x_t) - \nabla f(z_t)) \odot \sqrt{\eta_{t-1}} \|^2 + \frac{\gamma \lambda_1^2}{2} \| \nabla f(x_t) \odot \sqrt{\eta_{t-1}} \|^2$$

$$+ \frac{\gamma \lambda_2^2}{2} \mathbb{E} \| \frac{1}{b} \sum_{i \in B} (\nabla f_i(x_t + \rho_t \frac{\nabla f(x_t)}{\| \nabla f(x_t) \|}) - \nabla f_i(x_t)) \odot \sqrt{\eta_{t-1}} \|^2$$

$$+ \frac{\gamma \lambda_3^2}{2} \mathbb{E} \| \frac{1}{b} \sum_{i \in B} (\nabla f_i(x_t + \rho_t \frac{\sum_{i \in B} \nabla f_i(x_t)}{\| \sum_{i \in B} \nabla f_i(x_t) \|}) - \nabla f_i(x_t + \rho_t \frac{\nabla f(x_t)}{\| \nabla f(x_t) \|}) \odot \sqrt{\eta_{t-1}} \|^2 \tag{84}$$

$$\leq \frac{\gamma}{2}(\frac{1}{\lambda_1^2} + \frac{1}{\lambda_2^2} + \frac{1}{\lambda_3^2}) \mathbb{E} \| (\nabla f(x_t) - \nabla f(z_t)) \odot \sqrt{\eta_{t-1}} \|^2 + \frac{\gamma \lambda_1^2}{2} \| \nabla f(x_t) \odot \sqrt{\eta_{t-1}} \|^2$$

$$+ \frac{\gamma \lambda_2^2 L^2 \rho_t^2}{2\epsilon} + \frac{2 \lambda_3^2 \gamma L^2 \rho_t^2}{\epsilon} \tag{85}$$

$$\leq \frac{\gamma L^2}{2\epsilon}(\frac{1}{\lambda_1^2} + \frac{1}{\lambda_2^2} + \frac{1}{\lambda_3^2}) \mathbb{E} \| z_t - x_t \|^2 + \frac{\gamma \lambda_1^2}{2} \| \nabla f(x_t) \odot \sqrt{\eta_{t-1}} \|^2$$

$$+ \frac{\gamma \lambda_2^2 L^2 \rho_t^2}{2\epsilon} + \frac{2 \lambda_3^2 \gamma L^2 \rho_t^2}{\epsilon} \tag{86}$$

$$= \frac{\gamma^3 L^2 \beta_1^2}{2\epsilon(1 - \beta_1)^2}(\frac{1}{\lambda_1^2} + \frac{1}{\lambda_2^2} + \frac{1}{\lambda_3^2}) \| m_{t-1} \odot \eta_t - 1 \|^2 + \frac{\gamma \lambda_1^2}{2} \| \nabla f(x_t) \odot \sqrt{\eta_{t-1}} \|^2$$

$$+ \frac{\gamma \lambda_2^2 L^2 \rho_t^2}{2\epsilon} + \frac{2 \lambda_3^2 \gamma L^2 \rho_t^2}{\epsilon} \tag{87}$$

$$\leq \frac{\gamma^3 L^2 \beta_1^2}{2\epsilon(1 - \beta_1)^2}(\frac{1}{\lambda_1^2} + \frac{1}{\lambda_2^2} + \frac{1}{\lambda_3^2}) \frac{d G_\infty^2}{\epsilon^2} + \frac{\gamma \lambda_1^2}{2} \| \nabla f(x_t) \odot \sqrt{\eta_{t-1}} \|^2 + \frac{\gamma L^2 \rho_t^2}{2\epsilon}(\lambda_2^2 + 4 \lambda_3^2). \tag{88}$$

$$\square$$

**Lemma B.10.** *For the term defined in the algorithm, we have*

$$\frac{L}{2} \mathbb{E} \| z_{t+1} - z_t \|^2 \leq \frac{L G^2 \gamma^2 \beta_1^2}{(1 - \beta_1)^2} \mathbb{E} \| \eta_t - \eta_{t-1} \|^2$$

$$+ \gamma_t^2 L(3 \frac{1 + \beta}{\beta \epsilon}(\mathbb{E} \| \nabla f(x_t) \odot \sqrt{\eta_{t-1}} \|^2 + \frac{L \rho_t^2}{\epsilon} + \frac{\sigma^2}{b\epsilon}) + (1 + \beta) G^2 \mathbb{E} \| \eta_t - \eta_{t-1} \|^2) \tag{89}$$

*Proof.*

$$\frac{L}{2} \mathbb{E} \| z_{t+1} - z_t \|^2$$

$$= \frac{L}{2} \mathbb{E} \| \frac{\gamma \beta_1}{1 - \beta_1} m_{t-1} \odot (\eta_t - \eta_{t-1}) - \gamma g_t \odot \eta_t \|^2 \tag{90}$$

$$\leq \frac{L \gamma^2 \beta_1^2}{(1 - \beta_1)^2} \mathbb{E} \| m_{t-1} \odot (\eta_t - \eta_{t-1}) \|^2 + L \mathbb{E} \| \frac{\gamma_t}{b} \sum (\nabla f_i(x_t + \rho_t \frac{s_t}{\| s_t \|})) \odot \eta_t \|^2 \tag{91}$$

$$\leq \frac{L G^2 \gamma^2 \beta_1^2}{(1 - \beta_1)^2} \mathbb{E} \| \eta_t - \eta_{t-1} \|^2 + L \mathbb{E} \| \frac{\gamma_t}{b} \sum (\nabla f_i(x_t + \rho_t \frac{s_t}{\| s_t \|})) \odot \eta_t \|^2 \tag{92}$$

$$= \gamma_t^2 L \mathbb{E} \| \frac{1}{b} \sum (\nabla f_i(x_t + \rho_t \frac{s_t}{\| s_t \|})) \odot \eta_{t-1} + \frac{1}{b} \sum (\nabla f_i(x_t + \rho_t \frac{s_t}{\| s_t \|})) \odot (\eta_t - \eta_{t-1}) \|^2$$

$$+ \frac{L G^2 \gamma^2 \beta_1^2}{(1 - \beta_1)^2} \mathbb{E} \| \eta_t - \eta_{t-1} \|^2 \tag{93}$$

$$\leq \frac{L G^2 \gamma^2 \beta_1^2}{(1 - \beta_1)^2} \mathbb{E} \| \eta_t - \eta_{t-1} \|^2 + \gamma_t^2 L((1 + \frac{1}{\beta}) \mathbb{E} \| \frac{1}{b} \sum (\nabla f_i(x_t + \rho_t \frac{s_t}{\| s_t \|})) \odot \eta_{t-1} \|^2$$

$$+ (1 + \beta) \mathbb{E} \| \frac{1}{b} \sum (\nabla f_i(x_t + \rho_t \frac{s_t}{\| s_t \|})) \odot (\eta_t - \eta_{t-1}) \|^2) \tag{94}$$

$$\leq \gamma_t^2 L((1 + \frac{1}{\beta}) \mathbb{E} \| \frac{1}{b} \sum (\nabla f_i(x_t + \rho_t \frac{s_t}{\| s_t \|})) \odot \eta_{t-1} \|^2 + (1 + \beta) G^2 \mathbb{E} \| \eta_t - \eta_{t-1} \|^2)$$

$$+ \frac{L G^2 \gamma^2 \beta_1^2}{(1 - \beta_1)^2} \mathbb{E} \| \eta_t - \eta_{t-1} \|^2 \tag{95}$$

$$\leq \gamma_t^2 L((1+\frac{1}{\beta})\mathbb{E}\|\frac{1}{b}\sum(\nabla f_i(x_t+\rho_t\frac{s_t}{\|s_t\|}))\odot\sqrt{\eta_{t-1}}\|^2\times\|\sqrt{\eta_{t-1}}\|_\infty^2$$

$$+(1+\beta)G^2\mathbb{E}\|\eta_t-\eta_{t-1}\|^2)+\frac{LG^2\gamma^2\beta_1^2}{(1-\beta_1)^2}\mathbb{E}\|\eta_t-\eta_{t-1}\|^2 \tag{96}$$

$$\leq \gamma_t^2 L(\frac{1+\beta}{\beta\epsilon}\mathbb{E}\|\frac{1}{b}\sum(\nabla f_i(x_t+\rho_t\frac{s_t}{\|s_t\|}))\odot\sqrt{\eta_{t-1}}\|^2+(1+\beta)G^2\mathbb{E}\|\eta_t-\eta_{t-1}\|^2)$$

$$+\frac{LG^2\gamma^2\beta_1^2}{(1-\beta_1)^2}\mathbb{E}\|\eta_t-\eta_{t-1}\|^2 \tag{97}$$

$$\leq \gamma_t^2 L(3\frac{1+\beta}{\beta\epsilon}\mathbb{E}(\|\nabla f(x_t)\odot\sqrt{\eta_{t-1}}\|^2+\|(\frac{1}{b}\sum\nabla f_i(x_t)-\nabla f(x_t))\odot\sqrt{\eta_{t-1}}\|^2$$

$$+\|\frac{1}{b}\sum(\nabla f_i(x_t+\rho_t\frac{\sum_{i\in B}\nabla f_i(x_t)}{\|\sum_{i\in B}\nabla f_i(x_t)\|})-\nabla f_i(x_t))\odot\sqrt{\eta_{t-1}}\|^2)+(1+\beta)G^2\mathbb{E}\|\eta_t-\eta_{t-1}\|^2)$$

$$+\frac{LG^2\gamma^2\beta_1^2}{(1-\beta_1)^2}\mathbb{E}\|\eta_t-\eta_{t-1}\|^2 \tag{98}$$

$$\leq \gamma_t^2 L(3\frac{1+\beta}{\beta\epsilon}(\mathbb{E}\|\nabla f(x_t)\odot\sqrt{\eta_{t-1}}\|^2+\mathbb{E}\|\frac{1}{b}\sum(\nabla f_i(x_t+\rho_t\frac{\sum_{i\in B}\nabla f_i(x_t)}{\|\sum_{i\in B}\nabla f_i(x_t)\|})-\nabla f_i(x_t))\odot\sqrt{\eta_{t-1}}\|^2$$

$$+\frac{\sigma^2}{b\epsilon})+(1+\beta)G^2\mathbb{E}\|\eta_t-\eta_{t-1}\|^2)+\frac{LG^2\gamma^2\beta_1^2}{(1-\beta_1)^2}\mathbb{E}\|\eta_t-\eta_{t-1}\|^2 \tag{99}$$

$$\leq \gamma_t^2 L(3\frac{1+\beta}{\beta\epsilon}(\mathbb{E}\|\nabla f(x_t)\odot\sqrt{\eta_{t-1}}\|^2+\frac{1}{\epsilon}\mathbb{E}\|\frac{1}{b}\sum(\nabla f_i(x_t+\rho_t\frac{\sum_{i\in B}\nabla f_i(x_t)}{\|\sum_{i\in B}\nabla f_i(x_t)\|})-\nabla f_i(x_t))\|^2$$

$$+\frac{\sigma^2}{b\epsilon})+(1+\beta)G^2\mathbb{E}\|\eta_t-\eta_{t-1}\|^2)+\frac{LG^2\gamma^2\beta_1^2}{(1-\beta_1)^2}\mathbb{E}\|\eta_t-\eta_{t-1}\|^2 \tag{100}$$

$$\leq \gamma_t^2 L(3\frac{1+\beta}{\beta\epsilon}(\mathbb{E}\|\nabla f(x_t)\odot\sqrt{\eta_{t-1}}\|^2+\frac{1}{\epsilon b}\mathbb{E}\sum\|\nabla f_i(x_t+\rho_t\frac{\sum_{i\in B}\nabla f_i(x_t)}{\|\sum_{i\in B}\nabla f_i(x_t)\|})-\nabla f_i(x_t)\|^2$$

$$+\frac{\sigma^2}{b\epsilon})+(1+\beta)G^2\mathbb{E}\|\eta_t-\eta_{t-1}\|^2)+\frac{LG^2\gamma^2\beta_1^2}{(1-\beta_1)^2}\mathbb{E}\|\eta_t-\eta_{t-1}\|^2 \tag{101}$$

$$\leq \gamma_t^2 L(3\frac{1+\beta}{\beta\epsilon}(\mathbb{E}\|\nabla f(x_t)\odot\sqrt{\eta_{t-1}}\|^2+\frac{L\rho_t^2}{\epsilon}+\frac{\sigma^2}{b\epsilon})+(1+\beta)G^2\mathbb{E}\|\eta_t-\eta_{t-1}\|^2)$$

$$+\frac{LG^2\gamma^2\beta_1^2}{(1-\beta_1)^2}\mathbb{E}\|\eta_t-\eta_{t-1}\|^2. \tag{102}$$

$\square$

# C ADDITIONAL EXPERIMENT ILLUSTRATIONS

## C.1 EXPERIMENT ILLUSTRATIONS

We conduct the experiments on the GLUE benchmark with AdaSAM, AMSGrad, SAM and SGD, respectively. The optimizers do not have the momentum part ($\beta_1 = 0$). As a supplement to Table 3 and Table 4, Figure 2 and Figure 3 show the detailed loss and evaluation metrics versus number of steps curves during training. The loss curve of AdaSAM decreases faster than SAM and SGD in all tasks, and it has a similar decreasing speed as the AMSGrad. The metric curve of AdaSAM and AMSGrad show that the adaptive learning rate method is better than SGD and SAM. And AdaSAM decrease as faster as the AMSGrad in all tasks.

## C.2 ABLATION STUDY

In this section, we conduct experiments to evaluate the impact of momentum ($\beta_1 = 0.9$) on different optimizers. We show the experiment results in Figure 4 and Figure 5, respectively. The results are also illustrated in Table 1 and Table 2. SAM with adaptive learning rate (AdaSAM with $\rho = 0.9$) converges as fast as AMSGrad. Both AdaSAM and AMSGrad are faster than SGD with momentum and SAM with momentum. The generalization ability of AdaSAM with momentum is much better than SAM with momentum in all tasks. Besides, AdaSAM is also better than AMSGrad in GLUE benchmark except for QNLI and QQP tasks.

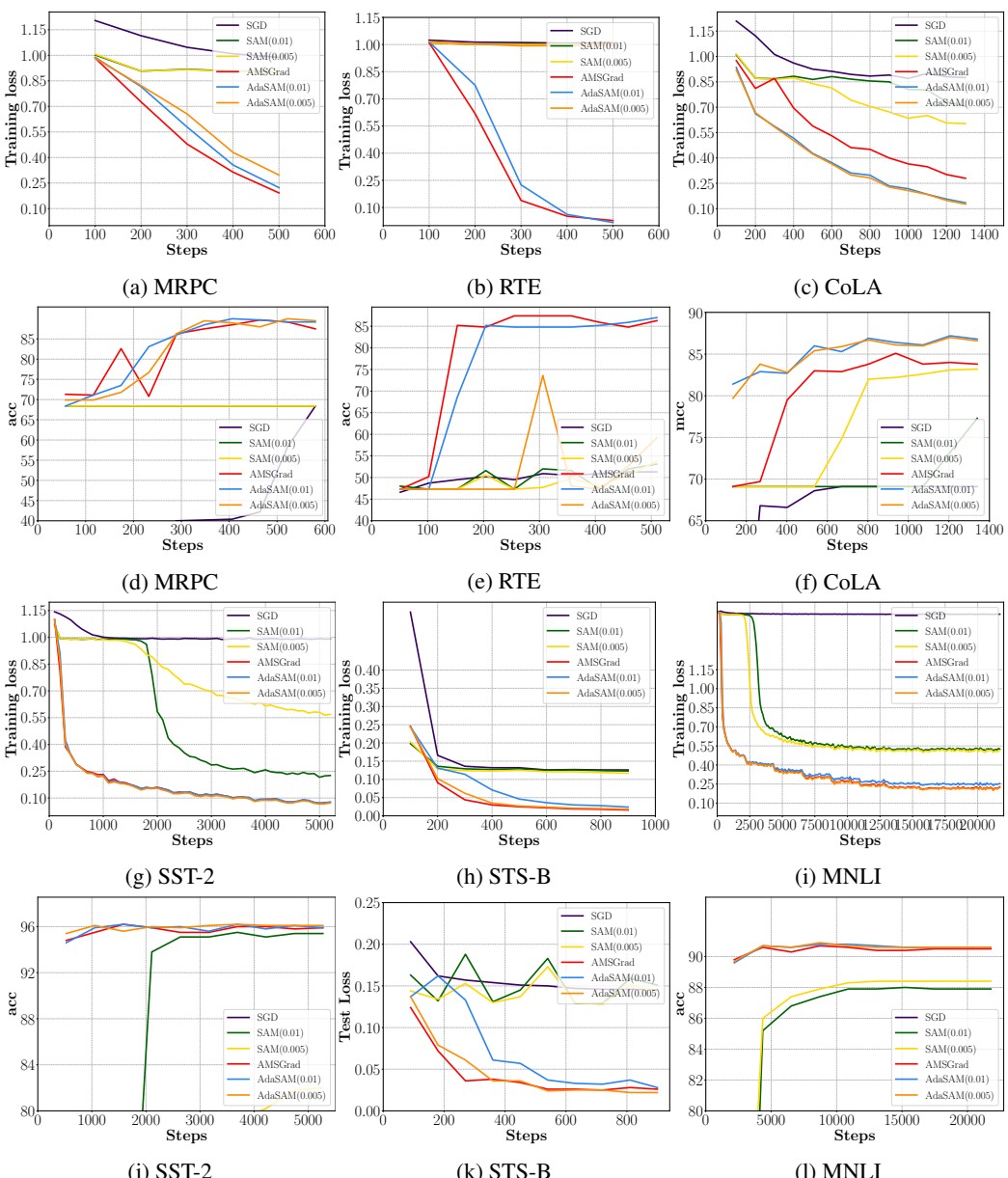

Figure 2: The loss and evaluation metric v.s. steps on MRPC, RTE, CoLA, SST-2, STS-B and MNLI.($\beta_1 = 0$)

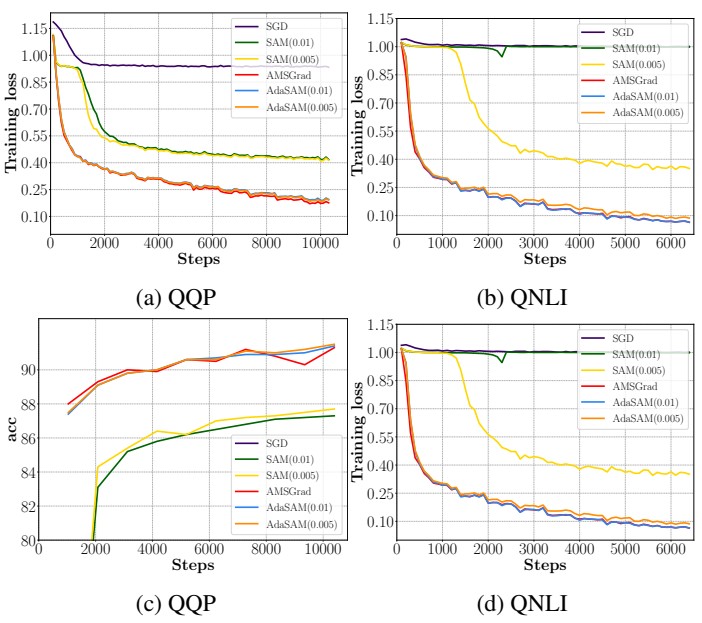

Figure 3: The loss and evaluation metric v.s. steps on QQP and QNLI.($\beta_1 = 0$)

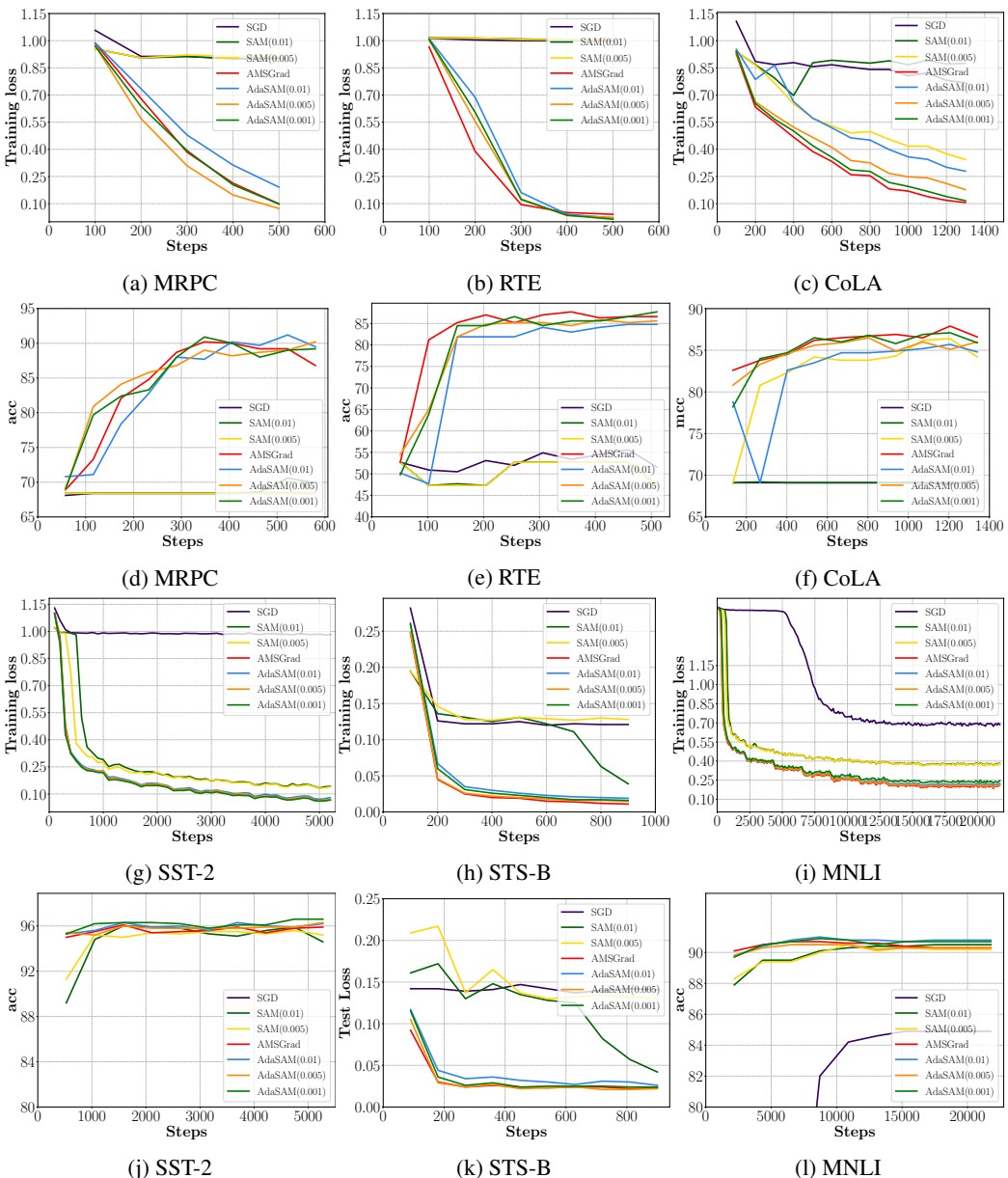

Figure 4: The loss and evaluation metric v.s. steps on MRPC, RTE, CoLA, SST-2, STS-B, MNLI.($\beta_1 = 0.9$)

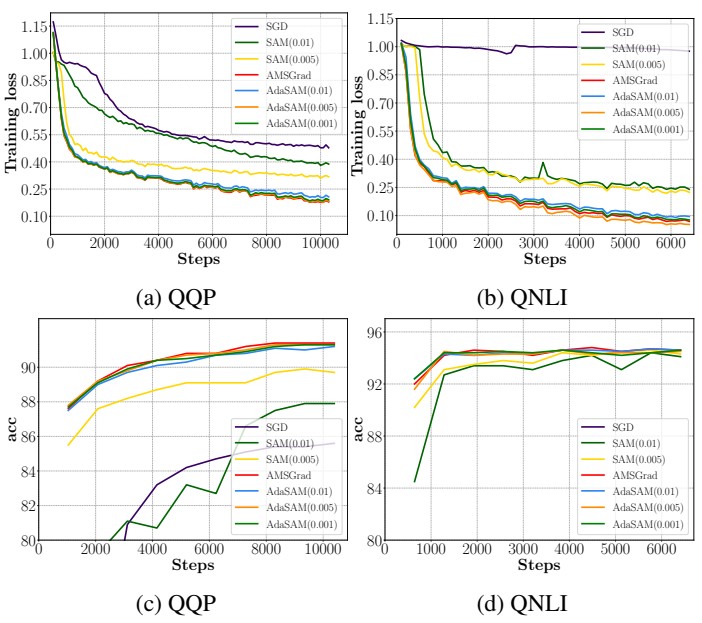

Figure 5: The loss and evaluation metric v.s. steps on QQP, QNLI.($\beta_1 = 0.9$)

