# OpenReview forum: "Provable Sharpness-Aware Minimization with Adaptive Learning Rate "
_ICLR.cc/2023/Conference — Submitted to ICLR 2023_

### Official Review · Reviewer_w96Y · 2022-10-22

**Confidence:** 4
**Clarity, Quality, Novelty And Reproducibility:** 1. clarity
**Correctness:** 4
**Technical Novelty And Significance:** 1
**Empirical Novelty And Significance:** 1
**Recommendation:** 3

**Strength And Weaknesses:**

The main weakness is that the theoretical results are not well motivated. As can be seen from equation (7) in Corollary 3.7. the main claimed contribution is that the algorithm converges (in expectation) to a first order stationary point of the smooth function $f$ which is defined in equation (2) and is a simple empirical risk minimization problem. The issue is that the goal of Sharpness aware minimization is not to minimize the original function but rather the regularized objective that is presented in equation (3). That is, if we wanted to achieve convergence of the form stated in equation (7) then we already have good algorithms for it: plain SGD on the objective $\sum_i f_i(x)$.

This is not surprising as the "convergence guarantee" is given when the regularization parameter $\rho$ tends to zero. But in this case what is the point? in this case the algorithm wouldn't be performing sharpness aware minimization at all!

A real interesting question is to obtain first-order optimality of equation (3) or equation (4) which are the real and approximated version of the sharpness aware minimization objective. Of course in this case the argument is much more difficult because (3) and (4) are non-smooth and non-convex objectives given the presence of a min-max structure in the case of (3), and the presence of a $\| \nabla f(x)\|$ in a denominator, in the case of (4). First order optimality guarantees in this case would involve more difficult concepts like the Clarke subdifferential.

When $\rho$ does not tend to zero equation (5) simply states that the algorithm converges to some vicinity of a first order stationary point of the original unregularized function $f$, which might not be interesting or useful. Overall, I believe that the authors are getting convergence rates for "something" for the sole purpose of showing some rate, but it is not an interesting problem to get a gurantee like (7) which, again, is already obtained by a simple algorithm like SGD.

The main strength is that the experiments appear to be extensive and the algorithm shows some practical improvements. However this is all eclipsed by the fact that the convergence analysis is presented as the main contribution.

**Summary Of The Paper:**

The authors propose an adaptive algorithm for sharpness aware minimization, a special type of regularized loss function that has a min-max structure. The adaptive algorithm is reminiscent of Adam. A convergence analysis to a first-order stationary value of the original unregularized loss is presented. The convergence of course requires that the size of the perturbation $\rho$ decreases to zero, which is weird because it means that no regularization is applied, but is consistent with the fact that the convergence is stated with respect to the unregularized loss. Experiments are presented showing that the algorithm can improve over some baselines.

**Summary Of The Review:**

The problem solved is not interesting because it is already solved by SGD (guarantees of the form of equation (7)). The algorithm seems to perform well in experiments but the theoretical results derived have no relation to the problem studied (sharpness aware minimization)

---

### Official Review · Reviewer_x2Pp · 2022-10-24

**Confidence:** 3
**Correctness:** 3
**Technical Novelty And Significance:** 2
**Empirical Novelty And Significance:** 2
**Recommendation:** 5

**Clarity, Quality, Novelty And Reproducibility:**

quality: the paper studies a natural extension of SAM and has some nice theory (though i have not read through all proofs due to (extremely) tight time constraint). however, there are mismatches between the theoretical and empirical results, and empirical results can be improved in terms of including additional tasks and baselines to make a stronger argument.
clarity: the paper is mostly well-written. authors should distinguish between \cite and \citep, and use the most appropriate one depending on the context.
novelty: the high-level idea of introducing adaptivity into SAM in new, to the best of my knowledge.
reproducibility: the experiments are likely reproducible.

**Strength And Weaknesses:**

strength: the paper studies an extension of SAM which to the best of my knowledge has not been studied before (although the idea seems natural, given precursors like Adam, RMSProp, and AdaGrad).

weaknesses:
- the paper shows limited empirical results regarding AdaSAM's advantage compared to optimizers (the intro claims that "The results show that AdaSAM outperforms the most of state-of-art optimizers"). more concretely, i think that to strengthen the empirical advantage argument, authors could try showing that
  - AdaSAM is competitive for tasks in other domains (e.g., the simplest image classification)
  - AdaSAM has an advantage over more "state-of-the-art" baselines (e.g., ASAM [1])
- a discrepancy between the theory and experiments: the theoretical result is about training-time convergence rate, whereas the it seems the results with GLUE are about test-time generalization performance (i wasn't able to pinpoint where the latter point was made, but i'd assume the numbers are from evaluation on the test spilt; unless i'm missing something obvious, authors should also clarify in the main text)

[1] Kwon, Jungmin, et al. "Asam: Adaptive sharpness-aware minimization for scale-invariant learning of deep neural networks." International Conference on Machine Learning. PMLR, 2021.


**Summary Of The Paper:**

the paper proposes the modification of AdaSAM to SAM (sharpness-aware minimization) that records running averages of first and second moment (based on the "SAM gradient"), and adjusts the parameter updated based on these estimates a la Adam and AdaGrad. authors show improved results with AdaSAM on GLUE benchmarks.

**Summary Of The Review:**

the paper proposes the modification of AdaSAM to SAM (sharpness-aware minimization) that records running averages of first and second moment (based on the "SAM gradient"), and adjusts the parameter updated based on these estimates a la Adam and AdaGrad. authors show improved results with AdaSAM on GLUE benchmarks. the work has some nice theory, but there are mismatches between the theoretical and empirical results, and empirical results can be improved in terms of including additional tasks and baselines to make a stronger argument.

---

### Official Review · Reviewer_5Rfi · 2022-10-25

**Confidence:** 3
**Correctness:** 2
**Technical Novelty And Significance:** 4
**Empirical Novelty And Significance:** 3
**Recommendation:** 5

**Clarity, Quality, Novelty And Reproducibility:**

## Clarity: The paper is overall clear, though some phrasing could be improved:
- "them to make them independent while taking an expectation. Then we bound them" the usage of them is slightly ambiguous and repetitive.
- "the learning rate tuning unbearable" seems to be an overstatement.
- ". Although remarkable performance has been achieved, their convergence is still unknown since the adaptive learning rate is used in SAM" What kind of adaptive learning rate are you talking about? The adaptive learning rate proposed in this paper does not seem to cover the one proposed in previous work.
- Corollary 3: you could put everything in the same $O()$, e.g. $O(1/\sqrt{bT} + 1/T)$
It remains minor, but I would like to point out that it would improve the paper.
## Quality:
- the quality of the theory seems to be above the ICLR threshold.
- to the extent of my knowledge, the result is novel.
- Regarding the experiments, there is, in my opinion a lot of room for improvement:
  1. error bars (or any estimation of the uncertainty) are missing. We do not know if the improvement over the baseline is statistically significant.
  2. It seems that the four tables are regarding the **same set of experiments**. It is very hard to compare the results going from one table to another. Moreover, it seems that some results are inconsistent:
    - The results of AdaSAM in Table 1 and Table 2 are different while they are supposed to report the same experiment.
    - The description of the experiments mentions that "Table 2 shows the result of SGD and SAM." but **neither** SGD nor SAM are reported in that table.
    - There is no experimental description of how the hyperparameters of SGD and AdaSAM are tuned. Thus one can wonder if the performance of SGD could be improved by tuning its hyperparameters.
3. Assuming that the HPs have been tuned fairly since there is no comparison with Adam or SAM, It is not clear whether AdaSAM is performing better because of the adaptivity (ADAM) or the sharpness awareness (SAM).
4. This work should compare AdaSAM with the other concurrent work on SAM (GSAM and ASAM).
5. In practice, even after the introduction of AMSGrad it is Adam that is used in practice. I think it would be worth comparing AdaSAM with and without Line 8 (in Alg 1).

## Reproducibility:
Because of the lack of experimental description and the lack of error bars and I am concerned about the reproducibility of the experiments in this paper.


**Strength And Weaknesses:**

## Strength:
- The theoretical analysis seems sound.
- The story and the intention are clear.
- The experiments are done with large-scale models.
## Weaknesses:
- I find the experimental could be significantly improved. Some results seems inconsistent and some comparisons are missing. Overall it is not clear if the comparison has been done fairly and whether adaptivity really improves SAM. (see more details in the reproducibility section)
- Many sentences are relatively ambiguous. It makes reading the paper more difficult than it could.


**Summary Of The Paper:**

This paper proposes an algorithm that combines adaptive step size (ADAM/AMSGrad) and sharpness-aware minimization (SAM). The author first show the convergence of this algorithm under standard assumptions (Lipschitz functions, finite variance, and uniformly bounded stochastic gradients.) Finally, they try their algorithm experimentally and compare it with some baselines.

**Summary Of The Review:**

Overall. I find the theoretical contribution significant but the experimental validation is under ICLR standards. Since Adam is mainly used because of its empirical success, it is essential to show that AdaSAM can be efficiently used in practice. Because the experimental section could be significantly improved, I consider this work marginally below the acceptance threshold. (this justify my current rating on correctness)

---

### Official Review · Reviewer_rzri · 2022-10-26

**Confidence:** 5
**Correctness:** 3
**Technical Novelty And Significance:** 1
**Empirical Novelty And Significance:** 1
**Recommendation:** 3

**Clarity, Quality, Novelty And Reproducibility:**

Clarity:
The paper lacks clarification for the meaning of "adaptive learning rate" and where the adaptiveness is coming from. The paper also did not explain the choice of many constants such as $\epsilon, \beta_1, \beta_2$. Please check the weakness section (W3).


Quality and Novelty:
The quality and novelty are limited. Please check the weakness section (W1, W2, W4).

Originality:
The paper is original, to the best of my knowledge.

**Strength And Weaknesses:**

Strength:


The paper is well-motivated due to the good empirical performance of SAM in the deep learning applications, and its extension to adaptive learning rate is an interesting topic.

Weaknesses:

W1: The main theoretical results (Theorem 3.6 and Corollary 3.7) are nothing surprising. It is easy to get the same convergence rate by simply running SGD on the original function $f$ but without a bounded gradient assumption (Assumption 3.4 is not required for SGD). In addition, Corollary 3.7 only considers a small radius $\rho=O(\sqrt{1/bT})$, which makes the proof extremely easy because the $\ell_2$ norm of the gradient of the original function and the gradient of the perturbed function can be bounded by $\rho L$ due to $L$-smoothness.

W2: The technical contribution of this paper is unclear to me. I do not see any technical difficulty by extending the analysis from SGD to SAM. It is also unclear to me what is the concrete meaning of adaptive learning rate in this paper. It seems to be that the algorithm still requires knowledge of the problem-dependent parameters such as the variance and the gradient upper bound, at least from formula (5) in Theorem 3.6.

W3: The paper lacks clarity. I do not understand what is the meaning of $\epsilon$ in the statement of Theorem 3.6 (I guess this is the missing constant in the denominator of the line 9 of the Algorithm 1 to make $\eta_t$ upper bounded by $1/\epsilon$). The effects of $\beta_1$ and $\beta_2$ are not explicitly considered in the statement or the proof.

W4: Experiments are rather weak. The authors did not compare best-tuned SAM versus AdaSAM directly. The paper also did not compare with adaptive variant such as Asam [Kwon et al. ICML 2021].


**Summary Of The Paper:**

This paper considered an adaptive variant of Sharpness Aware Minimization (AdaSAM) and theoretically showed the $O(1/\sqrt{bT})$ convergence rate and linear speedup property with respect to minibatch size $b$. The authors conducted several experiments including GLUE and showed that the AdaSAM with the best-tuned $\rho$ outperforms Amsgrad and SGD.

**Summary Of The Review:**

The paper studied an adaptive variant of SAM with corresponding complexity analysis. The technical contribution is unclear and the experiment is weak. I vote for rejecting the paper in its current form.

---

### Decision · Program_Chairs · 2023-01-20

**Decision:**

Reject

**Justification For Why Not Higher Score:**

Clear reject

**Justification For Why Not Lower Score:**

N/A

**Metareview: Summary, Strengths And Weaknesses:**

The reviewers found the theory presented in the submission to be severely lacking and straightforward. Given that the theory is the main contribution of the paper and the lack of a rebuttal from the authors, it is clear that the paper cannot be accepted to ICLR.